# Molecular simulation-based investigation of thiazole derivatives as potential LasR inhibitors of *Pseudomonas aeruginosa*

**Snigdha Bhardwaj[1], Kandasamy Nagarajan[2], Halima Mustafa Elagib[3], Sadaf Anwar[4], Mohammad Zeeshan Najm[5], Tulika Bhardwaj[6], Mohd Adnan Kausar[4]***

**1** KIET School of Pharmacy, KIET Group of Institutions, Ghaziabad, Uttar Pradesh, India, **2** Geomicrobiology Group, Department of Biological Sciences, University of Calgary, Calgary, Alberta, Canada, **3** Department of Pharmacology, College of Medicine, University of Ha'il, Hail, Saudi Arabia, **4** Department of Biochemistry, College of Medicine, University of Ha'il, Hail, Saudi Arabia, **5** School of Biosciences, Apeejay Stya University, Gurugram, Haryana, India, **6** Department of Agricultural, Food and Nutritional Sciences, University of Alberta, Edmonton, Alberta, Canada

* adnankausar1@gmail.com, ma.kausar@uoh.edu.sa

## Abstract

*Pseudomonas aeruginosa (P. aeruginosa)*, a very resilient pathogen, demonstrates a diverse array of virulence factors, the expression of which is closely linked to the quorum sensing(QS) mechanism, which facilitates cell-cell interaction. Quorum sensing (QS) inhibition is a promising strategy for combating bacterial infections. LasR, a transcriptional factor that controls the mechanism of QS in *P. aeruginosa*, is a promising target for therapeutic development, because a lot of research has been done on its structure. It has already been established that thiazoles and their compounds have anti-QS potential against P aeruginosa. The study aims to identify new LasR quorum sensing inhibitors (QSIs) derived from novel thiazoles utilizing a structure-based virtual screening technique using the ZINC database. A complete set of 800 molecules (a novel thiazole derivative library) were docked inside the active region of the LasR receptor before being screened using pharmacokinetic and toxicology studies. Among the derivatives that were examined, compounds D_152, D_153, and L_331 were selected as potential inhibitors of LasR in *P. aeruginosa* and further studied to obtain a crucial understanding of the binding interactions that take place between inhibitor ligands and LasR. The findings indicated that the pharmacophoric characteristics of the derivative D_152 were comparable to those of the reference thiazole molecule (TC). Moreover, the molecular docking investigations showed that derivative D_152 and reference compound TC both fit the LasR protein's active area well. Furthermore, TC and D_152's amino acid interaction graphs with LasR and CviR are nearly identical. Furthermore, compound D_152's ability to engage with the LasR binding site through the dissolution of the protein's dimer was demonstrated by molecular dynamics modeling tests conducted over a 50 ns time span, demonstrating its function as a LasR antagonist. Additionally, Density Functional Theory (DFT) study was conducted on compound D_152 in order to determine the electron density of a molecule. According to the research findings, the recently produced thiazole derivative (D_152) has the potential

**Data availability statement:** All relevant data are within the paper.

**Funding:** This research has been funded by Deputy for Research & Innovation, Ministry of Education through Initiative of Institutional Funding at University of Ha'il – Saudi Arabia through project number IFP-22 103. The funders had no role in study design, data collection and analysis, decision to publish, or preparation of the manuscript.

**Competing interests:** The authors have declared that no competing interests exist.

to be used as a QSI against the LasR receptor, which would speed up the fight against the pathogenicity of *P. aeruginosa* that is resistant to multiple drugs.

## 1. Introduction

Gram-negative, rod-shaped *P. aeruginosa* is an opportunistic, multidrug-resistant bacteria that can cause serious infections in humans. The bacteria is known for its inherent and sophisticated antimicrobial resistance mechanisms, which lead to severe hospital-acquired infections, particularly those with weakened immune systems and pneumonia related with ventilator use [1,2]. *P. aeruginosa* demonstrates a varied range of virulence factors, the expression of which is contingent upon the quorum sensing (QS) system, encompassing intercellular communication. The QS mechanism in *P. aeruginosa* is of significant importance in the dissemination of infections. It achieves this by regulating the formation of biofilms, secreting a diverse range of pathogenic agents, and facilitating the exchange of genetic material (DNA) [3–6]. The proliferation of bacterial biofilm occurs as it adheres to surfaces that are concealed by exopolymers, resulting in the formation of several layers of bacterial communities. Biofilm formation is a crucial factor in the development of secondary infections, which pose considerable challenges in terms of eradication. Consequently, bacteria develop resistance to traditional antimicrobial drugs [7,8]. In the QS mechanism, auto-inducers called N-acyl Homoserine Lactone (AHLs) are produced and released. At the appropriate concentrations, these AHLs form a receptor-AHL complex by binding to specific receptors. This complex facilitates the formation of biofilms and regulates the expression of pathogenic genes by acting as transcriptional regulators of key protein genes located in the QS region [9,10]. Presently, *P. aeruginosa* demonstrates the existence of a sophisticated QS network with four interconnected systems, such as las, rhl, iqs, and pqs. The four autoinducers that are used in these four QS pathways are N-(3-oxododecanoyl) -l-HSL (OdDHL), N-butanoyl-L-homoserine lactone (C4-HSL), 2-(2-hydroxyphenyl) -thiazole-4-carbaldehyde (IQS), and 2-heptyl-3-hydroxy-4(1H)-quinolone (PQS). The signaling molecules work through two distinct QS mechanisms: rhlI/R and lasI/R. N-(3-oxododecanoyl)-l-homoserine lactone (3-oxo-C12-HSL) is synthesized and perceived by LasI/R, while N-butanoyl-l-homoserine lactone (C4-HSL) is engaged in rhlI/R. LasR is triggered by the N-3-oxododecanoyl-L-homoserine lactone produced by lasI in response to stress (3-OXO-C12-HSL). LasR promotes the expression of lasA protease, lasB elastase, and pyoveridine, as well as the generation of pyrogen and biofilm formation. The transcriptional regulator rhlR reacts to N-butanoyl-L-homoserine lactone (C4 HSL), regulates rhamnolipid production, which is an essential component of the biofilm, and supervises the lasR transcriptional factor-activated genes [11,12]. The genes lasI and rhlI are accountable for the synthesis of AHL signaling molecules, while LasR and RhlR encode transcriptional activator proteins present in these circuits, indicating their genetic similarity [13,14]. Autoinducers, which are essential for the formation of bacterial biofilms and extracellular virulent elements, are of critical importance in the generation of intracellular signals [15,16]. Also implicated in the QS system of *P. aeruginosa* is the 2-heptyl-3-hydroxy-4-quinolone (PQS) signaling molecule. It modulates the lasB gene's expression., which is responsible for encoding the virulence-causing LasB elastases protein [17]. *P. aeruginosa* produces acute infections by three steps: adhesion, invasion, and systemic dissemination. It uses cell-associated and extracellular pathogenic elements to target host cells, causing skin damage and reducing immune system efficiency. The pathogen adheres to epithelial cells in immunocompromised patients and uses sugar-binding proteins like flagella, fimbriae (Polar, Type IV pili), and lectins (LecA and LecB) to produce elastases, LasA, and LasB

that cytotoxically affect respiratory cells and encourage bacterial adhesion to the mucosa of the airways. These enzymes included hydrolase elastin, an essential protein of connective tissue that is thought to play a crucial role in lung innate immunity. *P. aeruginosa* also stimulates the development of rhamnolipids and hemolytic phospholipases C, which are responsible for the breakdown of phospholipids (phosphatidylcholine) found in eukaryotic cell membranes and lungs. Additionally, the pathogen produces the redox toxin pyocyanin, which inhibits a variety of processes in mammalian cells, including metal-ion uptake and cell respiration. Following colonization at the infection site, the infection can spread throughout the body through systemic circulation using the same virulence factors involved in adhesion and invasion steps. This results in the formation of biofilms, a heterogeneous structure made up of extracellular DNA, proteins, rhamnolipids, and exopolysaccharide, at the colonized sites of host tissues with improved adhesion and stabilization. This leads to the establishment of a chronic infection and forms a physical barrier against various biocides, the immune system, UV light, and antimicrobial agents [18,19]. The authors suggest a novel methodology referred to as the "anti-virulence strategy" as a possible means of inhibiting the synthesis of virulent proteins and targeting signaling molecules. This approach aims to prevent bacterial infections without disrupting the proliferation of pathogens [20–22]. In the contemporary day, virtual screening has emerged as an indispensable component inside the drug discovery process. In-silico computational approaches provide significant benefits for the rapid identification and advancement of new small compounds as medications or leads. Moreover, these methodologies have the potential to effectively foretell the factors affecting the molecule's synergistic activity and its effectiveness in biological systems [23]. Molecular docking is a computer method that predicts how tiny molecules will interact to macromolecular targets in structure-based drug design. Bond conformations and free energies of binding are analyzed to help in the research of biomolecular interactions and the development of therapeutic drugs [24–26]. The SBVS method employs computational techniques to screen virtual libraries of compounds, enabling the assessment of an unlimited number of chemical structures *in silico*. Only a subset of these structures is chosen and subsequently subjected to laboratory screening studies [27,28]. The X-ray crystal structures of QS receptors such as LasR and RhlR have provided drug discovery scientists with the opportunity to find potential QS inhibitors utilizing the SBVS technique [29–31]. In their study, Kalia et al. employed the SBVS methodology to ascertain the QSIs targeting LasR. This was achieved by screening a total of 2603 ligands derived from natural sources, which were collected from the ZINC database. The SBVS strategy was utilized to find possible QSIs against the LasR master regulator in *P. aeruginosa* [32]. Thiazoles have diverse applications in drug development for treatment allergies, inflammation, HIV infections, hypertension, bacterial infections, hypnotics, schizophrenia, and pain, as novel inhibitors of bacterial DNA gyrase B, and as fibrinogen receptor antagonists with antithrombotic activity. They demonstrated excellent pharmacological activities such as antifungal, antibacterial, anti-inflammatory, analgesic, anti-cancer, and anticonvulsant properties. Substituted thiazole and benzothiazole compounds have previously been found to exhibit anti-QS activity against P aeruginosa [33–37]. There have been numerous reports of QS inhibitors from Acyl-homoserine lactones (AHL) analogues; the majority of them are members of the lactone, thiolactone, and furanone classes. Due to their (i) quick degradation in the presence of mammalian lactonases, (ii) strong sensitivity towards their cognate receptors, which renders these compounds autoinducers, (iii) potentially hazardous breakdown products, and other intrinsic limitations, these AHL analogues are not suitable for use as QS inhibitors. Considering the shortcomings of AHL analogues, Bharatam et al., (2015) studies unsymmetrical azines, a structurally distinct class of QS inhibitors and analyzed through the molecular docking of several thiazoles (unsymmetrical azines) against the LasR and Cvir receptors, followed by biological testing. Derivatives of azine

have been investigated for their molluscacidal, antifungal, antifilarial, anticancer, and opiate antagonist properties. Therefore, it can be said that the azine class of chemicals is safe to use in the creation of therapeutic candidates [38]. In this investigation, a well-established quantitative structure-invariant (QSI) chemical known as thiazole (TC) was employed as a template compound for the LasR protein of *P. aeruginosa*. Subsequently, a virtual screening method based on structural similarity was conducted. The ZINC database was utilized to search for structural similarities of the template compound (TC). Following the molecular docking process, the compounds that were screened underwent additional investigation to assess their in-silico pharmacokinetics and toxicity. The lead molecule that was found was also subjected to molecular dynamics simulations and density functional theory (DFT) analysis. The study is supported by the results of the molecular docking investigation as they reveal that these azines interact to the ligand binding region of the LasR protein. The thiazole molecule (an azine chromophore) established H-bonds and π-π interactions with the target protein. To investigate the stability of protein-ligand complexes, molecular dynamics simulations and DFT calculations were performed on the hit compound which is not being investigated yet. MD simulations confirm and validate the results of in-silico docking, revealing a reasonably excellent ADMET profile of the hit compound. Thus, thiazole compounds can be considered a hit for the development of possible quorum sensing inhibitors that could be utilized alone or in combination with established antibiotics to combat the pathogenicity of *P. aeruginosa*.

## 2. Material and methodology

### 2.1. Docking

**2.1.1. Data set preparation.** A reference molecule used was a thiazole compound (TC), specifically 2-((2-Chloroquinolin-3-yl)methylene)hydrazono)-3-methyl-2,3dihydrobenzo[d]thiaz, which has been confirmed to be a QSI for LasR of *P. aeruginosa*. The reference compounds' two-dimensional chemical structure was delineated and optimized geometrically using Chem Bio Draw ultra-version 12.0 2010, developed by Cambridge Soft, specifically Chem Bio office Ultra in 2010. The conformation of a molecule was stabilized through the utilization of the Molecular Mechanics 2 (MM2) force field, which facilitated geometric optimization and energy minimization. Once the energy minimization process was completed, all files were saved in mol2 format [39]. The chemical is present. In order to prepare the ligands, the mol2 format was loaded into AutoDock Vina v. 1.1.2. Fig 1 displays the reference molecule.

**2.1.2. Molecular docking technique for virtual screening.** This technique is a valuable tool for predicting the orientation of ligands within the crucial region of a protein, hence facilitating the identification of potential hits for a certain target [40]. In order to study possible therapeutic compounds based on their binding affinities and ligand-receptor interactions, it is beneficial to screen out extensive databases of chemicals against specific proteins or receptors [41]. The study was conducted by following a series of stages.

**Fig 1. Chemical structure of thiazole compound ($C_{18}H_{13}ClN_4S$) (2-((2-Chloroquinolin-3-yl)methylene)hydrazono)-3-methyl-2,3dihydrobenzo[d]thiaz).**

**2.1.3. Protein preparation.** The three-dimensional crystal structure of the transcriptional proteins CviR (PDB ID: 3QP5) and LasR (PDB ID: 2UV0), when intricated with the natural ligand Chlorolactone (HLC, MF: C14H16ClNO4) and 3-Oxo-N-[(3S)-tetrahydro-2-oxo-3-furanyl]-dodecanamide (OHN, MF: C16H27NO4) respectively, was obtained from the Protein Data Bank (PDB) repository and utilized for grid preparation [42]. Using Autodock Vina v. 1.1.2 software wizard, the protein was prepared for study [43]. Prior to the docking technique, several procedures were undertaken, including the elimination of water molecules, as they have the potential to impede the surface accessibility of the protein to the ligand. The docking procedures exclusively utilized chain E and its corresponding binding site, while disregarding any other homologous chains that may be present within the protein. The software tool assists in the incorporation of polar hydrogens, the allocation of partial atomic charges (Kollman charges), and the assignment of AD4 atoms to all atoms inside molecules [44–47].

**2.1.4. Ligand preparation.** During the process of ligand production, the AutoDock Vina software was utilized to incorporate several modules, such as hydrogen and Gasteiger charges. Subsequently, the AD4 atom type was assigned to each atom inside the ligand. To estimate the torsion angles of ligands across rotatable bonds, the roots were identified that results in various ligand conformations. The aromaticity criteria were then set to 7.5. Subsequently, the projected standard compounds underwent molecular docking investigations [48].

**2.1.5. Ligand docking.** To ensure confirmation, AutoDock Vina v. 1.1.2 was used to import only the E monomer of the Ligand Binding Domain (LBD) complexed with the OHN ligand in LasR (PDB ID: 2UV0) [43,49]. The water atoms were eliminated and the import stage utilized default values. The selection of the OHN ligand as the central point of the grid box dimensions was used to establish the docking grid. The reference ligand (known QSIs) was utilized for the initial molecular docking. The docking process did not involve the utilization of either tautomeric or alternate protonation states for the ligand. The energy scores for each molecule were obtained by docking them to the protein-ligand conformation, and a single docking posture was obtained after each run [50,51].

**2.1.6. Visualization.** The findings obtained by AutoDock Vina v. 1.1.2 [43,52] were analyzed using Pymol Software, specifically the academic version [53]. The software was utilized to generate three-dimensional images depicting the interaction between the protein and ligand combination.

## 2.2. Virtual screening based on structure similarity

Quorum sensing of high population density in *P. aeruginosa* triggers the generation of virulence components that could be lethal to the infected host. The signal 3-oxo-C12-HSL (N-3-oxododecanoyl-L-homoserine lactone), which is constitutively produced by *P. aeruginosa*'s LasI synthase, builds up with population increase and activates LasR, a transcriptional regulator (R protein) that is analogous to LuxR. Exoproteases, exotoxins, and secondary metabolites are among the numerous harmful virulence factors whose transcription is triggered by 3-oxoC12-HSL-induced LasR dimers binding target gene promoters. Additionally, biofilm maturation—which frequently leads to chronic pathogenic infections is facilitated by activated LasR. About thirty percent of the 350 genes that *P. aeruginosa* likely controls through quorum sensing encode virulence components [54]. So, the docking and ligand-protein interaction studies were performed for OHN and reported QSI molecule (TC) against target LasR protein (PDB ID: 2UV0) using AutoDock Vina v. 1.1.2. Based on the observations, the reference compound, TC (A azine derivative, proven QSI reported to *P. aeruginosa* LasR) was selected for structure based virtual screening using 3D structure similarity search of ZINC database (Lead Like Library and Drug Like Library) using SWISS Similarity approach [55,56].

Secondly, total 800 compounds were extracted from the ZINC database using Swiss similarity approach [55]. All the 800 compounds were docked using AutoDock Vina v. 1.1.2 to observe binding affinity scores (Kcal/mol) and ligand-protein interactions. Further, docked molecules which showed high scores and strong ligand-protein interactions were selected for *in-silico* ADMET studies [57]. The flowchart of methodology for identification of potential QSIs is depicted in Fig 2.

### 2.3. *In-silico* pharmacokinetic and toxicity (ADME/T) studies

Based on docking scores and ligand-protein interactions, the total number of 29 compounds (docking score ≥ -12.00 kcal/mol) was shortlisted and their protein-ligand interactions were predicted with LasR target protein (PDB ID: 2UV0). Further, these 29 compounds were evaluated for pharmacokinetic (ADME) and toxicity studies. We carried out computer aided ADME and Toxicity studies by using online PreADMET calculator program [58] to evaluate the pharmacokinetic and toxicity scores to shortlist the potential QSIs of *P. aeruginosa*. Some of the parameters that were calculated for ADME are computationally predicted Plasma protein binding (PPB) levels, blood brain barrier (BBB) level, human intestinal absorption (HIA) (Absorption and Caco-2 permeability) and for toxicity assessment (Ames test and Rodent carcinogenicity) were mainly examined [41,44].

### 2.4. Identification of potential QSIs

Based on docking score, interaction with LasR protein and literature support, among seven shortlisted LasR inhibitors, compounds D_152, D_153 and L_331 were identified as potential QSIs of LasR protein of *P. aeruginosa*.

### 2.5. Molecular Dynamics (MD) simulations

A minimization program was applied to the LasR-D_152 ligand complex using the DESMOND module of SCHRDINGER13. The system is neutralized by minimizing the complex structure within an orthorhombic box consisting of single point and charge (SPC) water and Cl- ions. The minimization of complex was performed using steepest descent and LBGFS vectors in a system builder until the energy difference between two consecutive stages reached

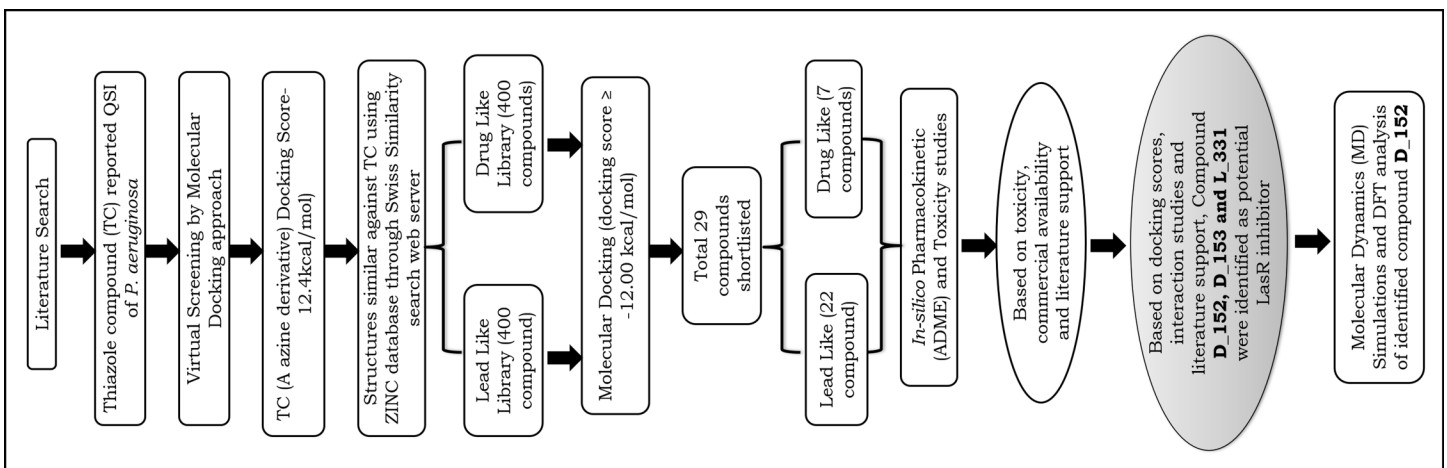

**Fig 2. Process flowchart used for identification of potential QSIs of LasR protein of *P. aeruginosa* using virtual screening and in-silico ADME/T studies.**

a maximum of 0.1 kcal/mol. The design in the DESMOND module of SCHRDINGER was used to minimize the complex in the MD simulations. Several parameters were established for the LasR-D_152 complex, including a simulation run length of 50 ns and wand frames with time steps of 50 ps. The experimental setup utilized a prototype device for molecular dynamics (MD) that resembles an orthorhombic box, but incorporates TIP3 water and Cl- ions. The simulations were conducted using a temperature of 300K and a pressure of 1.0325 bar. The equilibration of the system was achieved through the utilization of an NVT ensemble, which incorporated constant features such as particle counts, volume, and temperature. The frames obtained from the molecular dynamics (MD) runs were subjected to analysis utilizing the simulation interaction diagram tool in SCHRDINGER [59–62].

## 2.6. DFT calculations

Gaussian16 [63] was utilized to do quantum chemical computations. The B3LYP functional and 6-311 + G(d,p) were utilized for geometry optimizations. Subsequently, vibrational analysis was conducted to ascertain the absence of imaginary frequencies and to verify that the structures are situated at their minimum values. Energy estimates at the DFT/B3LYP/6-311 + G(d,p) level was also performed. GaussView6 [64] was used to view the structures, frontier molecular orbital (MO) surfaces, and molecular electrostatic maps (MESP).

## 3. Result and discussion

### 3.1. Molecular docking studies

*Chromobacterium violaceum (C. violaceum)*, namely the CviR/LuxR homologue, has been extensively employed as a prototype bacteria in the first evaluation of QSI for various gram-negative microorganisms, such as *P. aeruginosa* [65–67]. *C. violaceum* generates violacein, a QS-controlled purple pigment, and serves as a useful model for understanding the method of action of many traditional medicines. The QS system of this biomonitor strain is made up of the LuxI/LuxR homologue CviI/CviR. N-hexanoyl-L-acyl homoserine lactone (C6-AHL) and N-decanoyl-L-homoserine lactone (C10-HSL), which are autoinducer molecules made by autoinducer synthase CviI, are used by the CviI/CviR system to boost the expression of violacein synthesis, these molecules attach to the CviR (receptor protein). Molecular modeling was utilized because it can accurately predict the activity and binding affinities of possible QSI drugs against the protein active sites of test pathogens, *P. aeruginosa* and *C. violaceum* [68].

Hence, a molecular docking analysis was conducted to investigate the potential contacts established between the reference chemical and the CviR protein of *C. violaceum* and the LasR receptor of *P. aeruginosa*. The CviR protein (PDB ID:3QP5) of *C. violaceum* is captured in the x-ray crystal structure. Chen et al. (2011) reported the complexation of violaceum with the antagonist Chlorolactone (HLC, MF: C14H16CINO4) [69]. CviR is a homodimeric protein that consists of two binding units, namely LBD and the DNA Binding Domain (DBD), which are oriented in a hybrid manner. In a closed structural assembly, two DBD units are positioned at a considerable distance from each other and have a reduced affinity for DNA binding. The immediate conformation of the CviR protein is fixed by the antagonist HLC, resulting in the inhibition of QS action [70]. The interaction between HLC and the CviR receptor demonstrated that amide hydrogen is essential for the creation of hydrogen bonds. The HLC exhibited significant affinity towards specific amino acids of CviR, including Asp97, Trp84, and Ser155. These amino acids displayed hydrogen bonding, while Tyr88 and Leu72 displayed hydrophobic interactions (Fig 3). Conversely, the three-dimensional arrangement of LasR-LBDs displays symmetrical dimer configurations, in which the ligand is securely

incorporated within each monomer (Fig 4). LasR monomer exhibits a clubbed conformation, distinguished by the existence of three κ-helices facing a five-stranded β-sheet that is anti-parallel. The OHN ligand is oriented parallel to the β-sheet and α-helices (α3, α4, α5). The predominant intermolecular hydrogen bonding is facilitated by the α6-helix, whereas the β-sheet largely produces intermolecular hydrogen and hydrophobic bonds. The interactions between the dimers result in the formation of a significant surface area, which measures $1900^2 2$ [54]. The study on the re-docking of the native ligand (OHN) suggested that the software generated dependable predictions, consistent with findings in the existing literature. The molecular interaction between the natural ligand, OHN (AI), and the LasR protein is confirmed by the projecting AHL binding site in the structure of the LasR-LBD complex. The OHN ligand demonstrated hydrogen bonding interactions with all polar groups found in LasR-LBD, with the exception of the oxygen atom in the lactone ring. Fig 4 demonstrates that there are a total of four intermolecular interactions via hydrogen bond mainly involves Tyr-56, Trp-60, Ser-129, and Asp-73. The amino acids under consideration exhibit specificity and demonstrate distinct interactions in LuxR homologues, indicating that AHL and HSL may share a comparable activation mechanism owing to their identical functional groups. The close association of the OHN ligand with the LasR protein indicates a strong preference for this ligand and its detrimental impact on quorum sensing in various bacterial species[71,72].

The study was aimed to explore the binding affinity, docking pose, and molecular interactions of the reference chemical (TC) against the CviR (PDB ID: 3QP5) and LasR (PDB ID: 2UV0) receptors by molecular docking experiments. In comparison to their natural

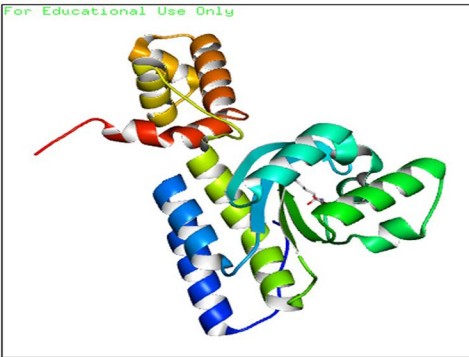

**Fig 3. The 3D cartoon representation of CviR (PDB ID:3QP5), co-crystallized with natural ligand Chlorolactone (HLC, MF: C14H16CINO4).**

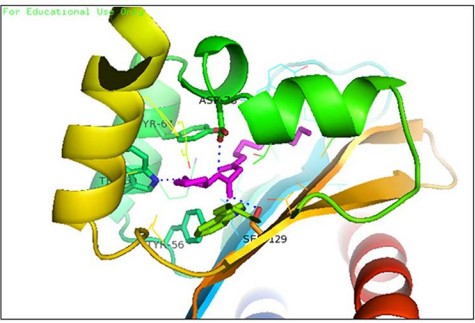

**Fig 4. The docking pose of co-crystallized standard ligand (OHN) with LasR (PDB ID: 2UV0).**

inhibitors/autoinducers, TC exhibited a higher binding affinity scores for both CviR and LasR proteins. Also, TC exhibited a greater binding affinity (-9.4 Kcal/mol) with the CviR receptor when compared to the natural inhibitor (HLC). Additionally, TC demonstrated a robust interaction with crucial amino acids, including Tyr88 and Trp84. Fig 5 displays the docking postures of TC with the CviR protein. The docking score of TC for the LasR receptor was determined to be -12.4 Kcal/moL, which was greater than the docking score of its natural ligand (OHN) which was -8.6 kcal/moL. The amino acid residues Ser129, Thr175, Trp60, and Tyr64 of LasR have been identified as the main binding sites for hydrogen bonding with TC, while Thr175, Trp60, and Tyr64 exhibit hydrophobic interactions (Fig 6). The study conducted by Chourasiya et al. (2015) and Bhardwaj et al. (2022) indicated that TC had comparable critical amino acid interaction with LasR [38,73]. TC, an azine derivative, is not an equivalent of OHN. However, the docking pose of TC exhibited a remarkable degree of overlap with OHN (Fig 5). Azines exhibit pharmacophoric characteristics akin to natural ligands through two mechanisms. Firstly, the benzothiazole nucleus in TC is enveloped by the hydrophobic acyl chain in OHN, leading to the formation of hydrophobic interactions. Secondly, the azine spacer (-C=N-N=C-) in TC coincides with the polar region, specifically the amide group (–CONH). The aromatic lactone ring in OHN is overspread by the aryl moiety of TC, resulting in its binding to the same location as the OHN lactone ring. Therefore, the chemical TC exhibits effective flexibility inside the active domain of the protein, thereby impeding the quorum sensing mechanism.

Many researches have shown that 'thiazoles derivatives' have a stronger attachment to the target protein (LasR) than the original ligand (OHN) [38,73–75]. However, it is important to note that TC does contain thiazole substitution. Hydrogen bonding is primarily responsible

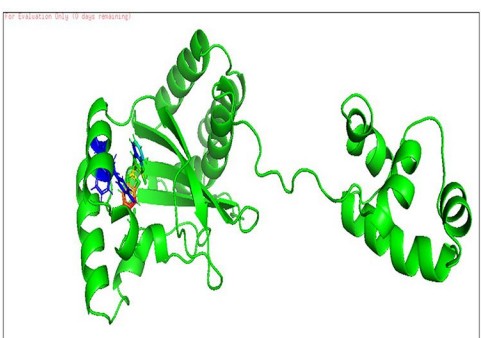

**Fig 5. Docking pose of reference Thiazole Compound (TC) inside active site pockets of CviR protein (3QP5).**

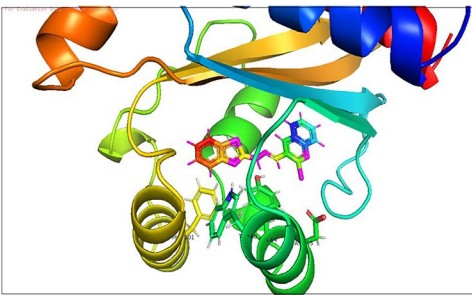

**Fig 6. Docking pose of reference Thiazole Compound (TC) inside active site pockets of LasR (2UV0).**

for the stability of the ligand-protein complex within the active target site of LasR protein. Another study highlighted the significance of hydrogen bonding and hydrophobic residues in preserving the docked-pose of azole compounds, emphasizing their relevance in ligand recognition at receptor sites [76]. The template structure chosen for the structure similarity search in the ZINC database was TC, which is a known QSI. The purpose of this search was to uncover new thiazole derivatives that are similar to LasR inhibitors of *P. aeruginosa*.

## 3.2. Virtual screening based on structure similarity

Virtual screening technique employed in this research study revealed diverse, yet specific ligand that binding in comparable manner similar to reference TC (known QSI) binding to LasR receptor [77]. Total 800 compounds were found to be structural similars' of TC which were obtained from search of Drug Like and Lead Like Libraries of ZINC database using Swiss Similarity Webserver. Total 800 molecules set was subsequently docked within the active pocket of LasR protein (PDB ID: 2UV0) using similar grid specification and docking protocol as employed for reference compound (TC) in AutoDock Vina v. 1.1.2 program. The docking score of the best pose of a reference chemical (a known inhibitor TC) is commonly employed as a cutoff criterion when selecting novel putative LasR inhibitors. If the best docking score of a structural similar compound is more negative than the docking score of the reference compound (TC), it is usually considered that the compound will be more active than the reference. The highest negative docking score values are typically used as a cutoff to choose a set of hit compounds and perform additional analysis on them when a reference molecule is not employed. Typically, the expected interactions between each chosen structure-similar chemical and the substrate-binding site of the LasR protein are examined [78]. The reference compound (TC) in this investigation has a docking score of -12.4 kcal/mol, indicating a higher anticipated binding to the LasR receptor. Therefore, in order to forecast the most powerful LasR inhibitors from the library searched structurally comparable compounds, the threshold value of the docking score was set at -12.0 kcal/mol and higher. Based on docking score, total 29 screened compounds were shortlisted and ranked (high to low based on docking scores) and protein-ligand interactions of the same set of compounds with LasR target protein (PDB ID: 2UV0) were predicted. The 29 compounds were further subjected to pharmacokinetic and toxicity studies. The total number of three test compounds were identified which exhibited favorable results for pharmacokinetic and toxicity parameters. Results of molecular docking along with ligand protein interaction data for all the potential 29 shortlisted compounds are given in Table 1.

## 3.3. *In-silico* pharmacokinetic and toxicity (ADME/T) studies

The pharmacokinetic and toxicological profiling (ADME/T) were done for all the 29 shortlisted compounds using online server PreADMET and SwissADME [37]. The program provides analysis on molecular fingerprinting of employed chemical structure resulting in generation of pharmacokinetic data based on specific chemical nature of a molecule. Significant work is devoted to the early estimation of PPB, primarily with respect to human serum albumin (HSA), the most prevalent and significant plasma protein, in holistic drug design techniques, where ADME(T) qualities are taken into account concurrently with target affinity. The structure-based approaches are helpful in locating the precise binding location and could be crucial for examining drug-drug interactions involving plasma protein binding (PPB) [79]. Plasma protein binding is an essential factor influencing the pharmacokinetic properties of chemical substances in living organisms. The information implied that almost all test compounds have good PPB efficiency (>90%). For the studied ligands, acute oral toxicity was determined to be minimal. Furthermore, the test compounds are not expected to cross

**Table 1. The details of shortlisted 29 test compounds with IUPAC names, ligand binding affinity (Kcal/mol) and ligand-protein interaction with LasR protein (PDB ID 2UV0) of shortlisted 29 test compounds.**

| S. No | Compound Code | ZINC/ Pub-chem ID | IUPAC Names | Structures | Docking Scores (Kcal/ mol) | Interaction with Amino acids of LasR (2UV0) | |
|---|---|---|---|---|---|---|---|
| | | | | | | Hydrogen Bond | Van der waals Interactions |
| 1 | D_152 | ZINC16650743 | 1-phenyl-4-(4-phenyl-1,3-thiazol-2-yl)-1H-1,2,3-triazol-5-amine | | −12.2 | Ser129 | Leu125,Ala127,Tyr64,Tyr47,Trp60,Phe101,Asp73 |
| 2 | D_153 | ZINC67334656 | 4-(1H-1,3-benzodiazol-2-yl)-1-phenyl-1H-1,2, 3-triazol-5-amine | | −12.3 | Ser129 | Trp88,Tyr93,Thr75,Tyr56,Tyr64,Ala127,Gly38,Leu36,Leu110,Asp73 |
| 3 | D_211 | 82370750 | 5-(4-ethylphenyl)-3-phenyltriazol-4-amine | | −12.8 | Ser129 | Trp88,Leu110,Tyr93,Thr75,Asp73,Ala127,Leu36,Gly38,Tyr64 |
| 4 | D_244 | ZINC38486684 | 1-(3-fluorophenyl)-4-(4-phenyl-1,3-thiazol-2-yl)-1H-1,2,3-tri-azol-5-amine | | −12.1 | – | Phe101,Val76,Aap73,Ala127,Leu36,Tyr64,Ser129,Trp88 |

*(Continued)*

**Table 1.** (Continued)

| S. No | Compound Code | ZINC/ Pub-chem ID | IUPAC Names | Structures | Docking Scores (Kcal/ mol) | Interaction with Amino acids of LasR (2UV0) | |
|---|---|---|---|---|---|---|---|
| | | | | | | Hydrogen Bond | Van der waals Interactions |
| 5 | D_270 | 29102955 | N-(6,7-dihydro-5H-pyrrolo[1,2-d][1,2,4]triazol-3-ylmethyl)-5-naphthalen-1-yl-1,2,4-triazin-3-amine |  | −13.6 | Ser129 | Ala127,Phe101,Leu36, Tyr56,Val76,Gly38, Trp60 |
| 6 | D_292 | ZINC57717 | 5-phenyl-3-(4-phenylpyridin-2-yl)-1,2,4-triazine |  | −13 | Ser129,Thr75,Thr115, Trp88 | Asp73,Gly38,Phe101, Leu36,Tyr64,Tyr56,Trp60,Leu110,Ala105 |
| 7 | D_295 | ZINC174021 | 2-[6-(1H-benzimidazol-2-yl)pyridin-2-yl]-1H-benzimidazole |  | −12.1 | – | Asp73,Val76,Ala127, Leu36,Tyr56,Tyr64,Trp60,Thr75 |
| 8 | L_12 | 82436949 | 5-[4-cyclopropyl-2-(2-methylpropyl)-1,3-thiazol-5-yl]-N-ethyl-1,3,4-thiadiazol-2-amine |  | −12.5 | Ser129 | Thr75,Trp88,Ala127, Val76,Leu36,Asp73,Tyr56,Leu110,Tyr64 |
| 9 | L_13 | ZINC67334656 | 5-(1H-benzimidazol-2-yl)-3-phenyltriazol-4-amine |  | −12.3 | Ser129 | Thr75,Tyr93,Leu36,Gly38,Asp73,Tyr56,Tyr64,Leu110 |
| 10 | L_27 | ZINC523912 | 3-(1H-benzimidazol-2-ylmethylsulfanyl)-5-(3-methylphenyl)-1,2,4-triazol-4-amine |  | −12 | Thr75 | Trp88,Asp73,Leu36,Val76,Gly38,Tyr64,Tyr56,Phe101,Gly126 |

*(Continued)*

**Table 1.** (Continued)

| S. No | Compound Code | ZINC/ Pub-chem ID | IUPAC Names | Structures | Docking Scores (Kcal/mol) | Interaction with Amino acids of LasR (2UV0) | |
|---|---|---|---|---|---|---|---|
| | | | | | | Hydrogen Bond | Van der waals Interactions |
| 11 | L_31 | ZINC523760 | 3-benzylsulfanyl-5-pyridin-2-yl-1,2,4-triazol-4-amine |  | −12.8 | Ser129 | Thr75,Asp73,Ala127, Gly38,Leu36,Tyr56,Tyr93,Trp88,Ala50 |
| 12 | L_40 | ZINC38486684 | 3-(3-fluorophenyl)-5-(4-phenyl-1,3-thiazol-2-yl)triazol-4-amine |  | −12.1 | – | Leu125,Val76,Ala127, Leu36,Asp73,Ser129, Phe101,Tyr56,Tyr64 |
| 13 | L_43 | 29102955 | N-(6,7-dihydro-5H-pyrrolo[2,1-c][1,2,4]triazol-3-ylmethyl)-5-naphthalen-1-yl-1,2,4-triazin-3-amine |  | −13.6 | Ser129 | Tyr47,Ala127,Leu36, Val76,Thr75,Trp88,Leu125,Tyr56,Tyr54, Phe101 |
| 14 | L_85 | ZINC65487672 | N-(2-methyl-1H-indol-5-yl)imid-azo[1,2-a]pyri-dine-2-carboxamide |  | −12.7 | Ser129 | Gly38,Ala127,Leu36, Trp88,Trp60,Tyr56,Ala105,Asp73,Ala50 |
| 15 | L_101 | ZINC9367895 | 2-[[1-(4-methylphenyl)tetrazol-5-yl]sulfanylmethyl]quinazolin-4-amine |  | −12.5 | Thr75 | Phe101,Tyr64,Tyr56, Thr115,Gly38,Gly126, Val76,Leu36,Arg61, Leu110 |

*(Continued)*

**Table 1.** (Continued)

| S. No | Compound Code | ZINC/ Pub-chem ID | IUPAC Names | Structures | Docking Scores (Kcal/mol) | Interaction with Amino acids of LasR (2UV0) | |
|---|---|---|---|---|---|---|---|
| | | | | | | Hydrogen Bond | Van der waals Interactions |
| 16 | L_132 | ZINC4618957 | 2-[(1-benzyl-1H-1,2,3-triazol-4-yl)methyl]-5-phenyl-2H-1,2,3,4-tetrazole | | −12.4 | Tyr56, Ser129 | Leu110,Trp88,Ala105, Thr75,Tyr64,Leu36,Gly126,Leu125,Val76 |
| 17 | L_151 | ZINC95467676 | 5-pyridin-2-yl-N-([1,2,4]triazolo[4,3-a]pyridin-3-ylmethyl)-1,3,4-thiadiazol-2-amine | | −12.1 | Ser129 | Tyr64,Trp88,Phe101,Gly126,Gly38, Leu110, Trp60,Leu36,Ala105 |
| 18 | L_154 | ZINC152694 | 1-phenyl-3-(quinolin-2-yl)urea | | −12.3 | – | Phe101,Trp88,Asp73, Gly38,Leu36,Tyr64,Tyr56,Leu110,Tyr93, Thr75 |
| 19 | L_175 | 3704385 | 1-[[3-(benzotriazol-1-ylmethyl)-1H-1,2,4-triazol-5-yl]methyl]benzotriazole | | −12.9 | Ser129 | Trp88,Thr75,Phe101, Gly38,Tyr47,Gly126, Leu125,Tyr64,Tyr58, Leu110 |
| 20 | L_195 | ZINC89296180 | 2-[(5-ethyl-4-methyl-1,2,4-triazol-3-yl)methyl]-5-(4-propan-2-ylphenyl)tetrazole | | −12.3 | Ser129, Thr75 | Val76,Tyr47,Leu36,Tyr56,Tyr64,Phe101, Ala127,Gly126,Trp88,Thr75 |

*(Continued)*

**Table 1.** (Continued)

| S. No | Compound Code | ZINC/ Pub-chem ID | IUPAC Names | Structures | Docking Scores (Kcal/ mol) | Interaction with Amino acids of LasR (2UV0) | |
|---|---|---|---|---|---|---|---|
| | | | | | | Hydrogen Bond | Van der waals Interactions |
| 21 | L_205 | ZINC2239455 | 3-(benzotriazol-1-ylmethyl)-6-benzyl-[1,2,4]triazolo[3,4-b][1,3,4]thiadiazole | | −12.5 | – | Ser129,Phe101,Ala127,Gly38,Tyr93,Leu36,Tyr64,Leu110,Trp88 |
| 22 | L_223 | ZINC31872616 | N-(2-chlorophenyl)-1-phenyl-1H-1,2,3-triazole-4-carboxamide | | −12.2 | Tyr56, Ser129 | Trp88,Ala105,Leu110,Tyr64,Ala127,Tyr47,Leu36,Tyr56,Val76,Asp73 |
| 23 | L_288 | 60385036 | N-(1H-benzimidazol-2-yl)-1H-indole-2-carboxamide | | −12.7 | Ser129, Thr75 | Asp73,Trp88,Tyr64,Ala127,Gly38,Tyr47,Leu110,Tyr56,Val76,Ala50 |
| 24 | L_293 | 60495701 | 1,5-Dimethyl-2-(2-phenyltriazol-4-yl)benzimidazole | | −12 | Ser129, Thr75 | Asp73,Phe101,Tyr64,Leu36,Tyr56,Trp88,Ile52,Val76,Tyr47 |
| 25 | L_316 | ZINC31804832 | N-{[1,2,4]Triazolo[4,3-a]pyridin-3-yl}quinoline-2-carboxamide | | −12.8 | Trp60, Ser129 | Asp73,Phe101,Tyr64,Tyr56,Ala127,Trp88,Leu110,Ala105,Leu36,Val76 |
| 26 | L_328 | ZINC49029776 | N-[(2-Chlorophenyl)methyl]-7-phenylpyrazolo[3,4-d]triazin-4-amine | | −12.1 | – | Ser129,Ala127,Phe101,Tyr64,Leu36,Gly126,Leu40,Val76,Trp60 |

*(Continued)*

**Table 1.** (Continued)

| S. No | Compound Code | ZINC/ Pub-chem ID | IUPAC Names | Structures | Docking Scores (Kcal/mol) | Interaction with Amino acids of LasR (2UV0) | |
|---|---|---|---|---|---|---|---|
| | | | | | | **Hydrogen Bond** | **Van der waals Interactions** |
| 27 | L_331 | ZINC32760540 | 3-methyl-N-[(3-methylphenyl)methyl]quinoline-8-sulfon-amide |  | −12.1 | Thr75 | Asp73,Val76,Ala127,Phe101,Tyr56,Asp73,Trp88,Tyr93,Leu125 |
| 28 | L_364 | ZINC13844131 | 1-{[(6-Phenyl-1,2,4-triazin-3-yl)sulfanyl]methyl}-1H-benzotriazole |  | −12.5 | Ser129, Thr75, Tyr56 | Phe101,Trp60,Asp73,Val76,Leu110,Leu36,Gly38,Ala127,Ala105 |
| 29 | L_386 | 136271159 | 2-(2-Phenylethyl)-6H-[1,2,4]triazolo[1,5-c]quinazolin-5-one |  | −12.2 | – | Ala127,Leu36,Val76,Trp88,Leu110,Ser110,Leu36,Arg61,Thr75 |

the blood-brain barrier, which may be an advantage since the CNS is not a target of interest for these compounds, and this could also prevent possible drawbacks related to it [80]. BBB penetration Log values were found to be significant i.e., below 1 for all the test compounds except L_154, L_288. Greater HIA indicates that the substance may be better absorbed from the intestinal system after oral ingestion. All the test compounds exhibit good oral absorption profile and scores were found under the 100% absorption ellipse. The Caco-2 cell permeability results indicate that majority of test compounds exhibited the desired values under range of 4-70% except D_153, L_14, L_132 and L_175. The AMES toxicity test is used to determine if a chemical is mutagenic. Test ligands that showed a negative AMES toxicity test result are regarded as non-mutagenic [81]. In toxicity studies, Ames test predicted the test compounds as mutagens with negative or positive outcome for rodent carcinogenicity (mouse and rats). Besides the efficient correlation with carcinogenicity, sometimes it is very difficult to interpret a positive inference [82]. The *in-silico* data of pharmacokinetic and toxicity studies for all 29 lead compounds are provided in Table 2.

### 3.4. Identification of potential QSIs

After analyzing the results obtained from *in-silico* docking and ADMET studies, the compounds with ZINC IDs: ZINC16650743 (D_152), ZINC67334656 (D_153), ZINC32760540

**Table 2. Details of pharmacokinetic and toxicity prediction of 29 test compounds using SwissADME.**

| Test Compounds | PPP | BBB | HIA | | Toxicity | |
|---|---|---|---|---|---|---|
| | | Log value | Absorption | Caco-2 cell permeability | Ames Test | Rodent Carcinogenicity (Mouse/Rat) |
| Reference Values | < 90% | < 1 | < 100% | 4 to 70% | Negative result | |
| D_152 | 89.51 | 0.7506 | 98.056792 | 18.3937 | Mutagen | Both Negative |
| D_153 | 88.621 | 0.9203 | 90.510671 | 2.69293 | Mutagen | Both Negative |
| D_211 | 90.164975 | 0.0552 | 96.122342 | 19.9678 | Mutagen | Both Positive |
| D_244 | 100 | 0.3195 | 98.05163 | 33.8639 | Mutagen | Mouse: Positive, Rat: Negative |
| D_270 | 93.85201 | 0.0497 | 96.609159 | 18.0884 | Mutagen | Mouse: Negative, Rat: Positive |
| D_292 | 100 | 3.7726 | 97.368145 | 46.5159 | Mutagen | Both Negative |
| D_295 | 93.066188 | 2.0567 | 92.059063 | 20.6205 | Mutagen | Both Negative |
| L_12 | 96.51667 | 0.750644 | 98.05679 | 18.3937 | Mutagen | Mouse: Positive, Rat: Negative |
| L_13 | 100 | 0.920327 | 90.51067 | 2.69293 | Mutagen | Mouse: Positive, Rat: Negative |
| L_27 | 94.03396 | 0.149091 | 92.35838 | 7.14106 | Mutagen | Both Negative |
| L_31 | 90.16498 | 0.055153 | 96.12234 | 19.9678 | Mutagen | Both Positive |
| L_40 | 100 | 0.279521 | 98.05163 | 24.4065 | NA | NA |
| L_43 | 93.85201 | 0.049669 | 96.60916 | 18.0884 | Mutagen | Mouse: Negative, Rat: Positive |
| L_85 | 84.63182 | 0.582387 | 91.93382 | 33.8903 | Mutagen | Both Positive |
| L_101 | 96.63645 | 0.182943 | 98.57931 | 18.9739 | Mutagen | Mouse: Negative, Rat: Positive |
| L_132 | 96.36002 | 0.291962 | 98.89119 | 0.840722 | Mutagen | Mouse: Negative, Rat: Positive |
| L_151 | 83.45483 | 0.089595 | 97.14564 | 6.0034 | Mutagen | Both Negative |
| L_154 | 100 | 1.90455 | 93.93035 | 21.4123 | Mutagen | Mouse: Positive, Rat: Negative |
| L_175 | 97.69898 | 0.170127 | 93.67204 | 0.758772 | Mutagen | Mouse: Negative, Rat: Positive |
| L_195 | 95.48222 | 0.238159 | 98.65282 | 4.24638 | Mutagen | Mouse: Negative, Rat: Positive |
| L_205 | 98.51081 | 0.502218 | 99.44309 | 7.32273 | Mutagen | Both Negative |
| L_223 | 98.08622 | 0.545112 | 95.93148 | 21.4833 | Mutagen | Both Negative |
| L_288 | 92.51538 | 1.14346 | 88.89051 | 16.9497 | Mutagen | Mouse: Positive, Rat: Negative |
| L_293 | 95.56938 | 3.54157 | 97.40322 | 39.6105 | Mutagen | Both Negative |
| L_316 | 100 | 0.394137 | 96.18608 | 20.8285 | Mutagen | Both Negative |
| L_328 | 100 | 0.214292 | 96.48775 | 21.7286 | Mutagen | Both Negative |
| L_331 | 98.256 | 0.808342 | 96.24944 | 18.6286 | Mutagen | Both Negative |
| L_364 | 98.30578 | 0.574538 | 99.39481 | 21.4986 | Mutagen | Both Negative |
| L_386 | 87.994 | 1.0384 | 96.40647 | 21.5407 | Mutagen | Both Negative |

*NA= Data Not Available.

(L_331) were identified as QSIs with potential antagonistic activity towards LasR protein of *P. aeruginosa*. Molecular docking analysis was performed for all the three identified compounds D_152, D_153 and L_331 to elucidate putative interaction with CviR and LasR receptors. The details of docking score and ligand-protein interaction of TC and D_152, D_153 and L_331 with CviR receptor is shown in Table 3. Data suggested strong interactions with key amino acids with these compounds. The docking poses of D_152, D_153 and L_331 with CviR receptor are shown in Figs 7–9 respectively and LasR receptor are shown in Figs 10–12 respectively.

### 3.5. Molecular Dynamics (MD) simulations

Molecular Dynamics Simulations examine the actual motions of molecules and atoms. These simulations provide a very fine temporal resolution and complete atomic detail of the behavior of proteins and other biomolecules, commonly referred to as ligands. The simulation's and

**Table 3. The details of selected hit compounds with ligand binding affinity (Kcal/mol) and ligand-protein interaction with CviR protein (PDB ID: 3QP5).**

| Identified Compounds | Docking Score (kcal/mol) | Ligand-Protein interactions (CviR) |
|---|---|---|
| D_152 | −8.674 | Asp 97, Tyr88, Trp84 |
| D_153 | −8.431 | Tyr88, Trp84, Ala127 |
| L_331 | −8.501 | Trp84, Asp73, Ser129 |

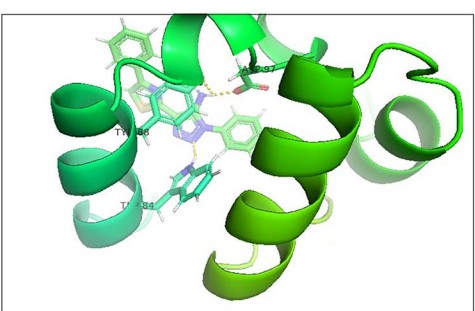

**Fig 7. Docking pose of compound D_152 inside active site pockets of CviR protein (3QP5).**

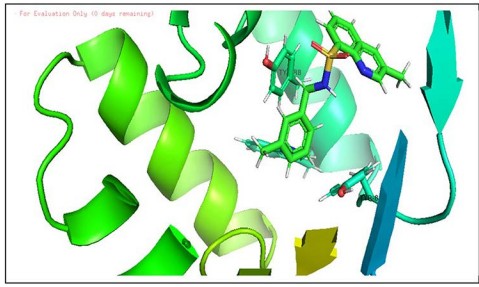

**Fig 8. Docking pose of compound D_153 inside active site pockets of CviR protein (3QP5).**

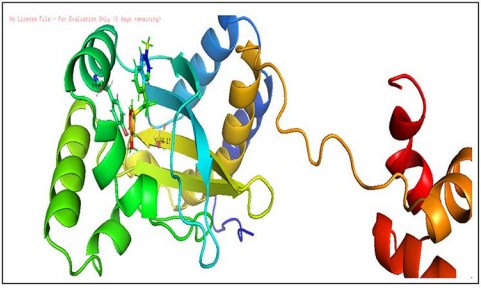

**Fig 9. Docking pose of compound L_331 inside active site pockets of CviR protein (3QP5).**

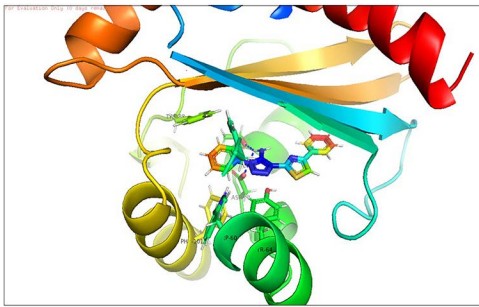

**Fig 10. Docking pose of compound D_152 inside active site pockets of LasR protein (2UV0).**

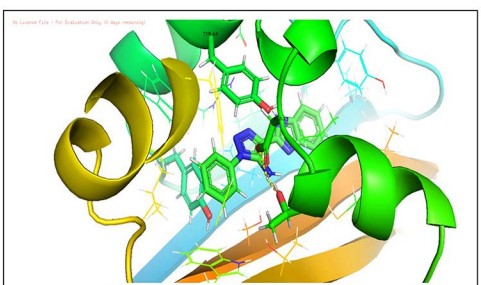

**Fig 11. Docking pose of compound D_153 inside active site pockets of LasR protein (2UV0).**

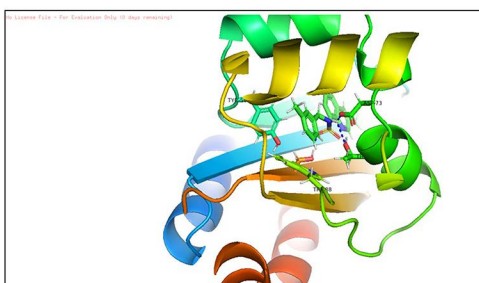

**Fig 12. Docking pose of compound L_331 inside active site pockets of LasR protein (2UV0).**

MD trajectory's duration is constrained by the time step. When energy (E) increases quickly over time, an MD simulation may become unstable due to excessively long timesteps. The TIP4P water environment was used for the simulation, and it was run at 10-ns intervals. The MD simulation of protein (LasR)-ligand (D_152) complex (LasR-D_152) was subjected to simulation for 10 ns [83].

Molecular dynamics provides deep knowledge about change in conformation of a protein-ligand complex as it explores hidden dynamic patterns of protein in a complex state which is associated with protein function [84–86]. The MD simulation of protein-ligand (LasR-D_152) complex was subjected to simulation for 10 ns. The modeling parameters such as energy (potential & total), pressure and temperature were set to stabilize the simulated complex [87]. Potential energy (the sum of bonded and non-bonded expressions)

of introduced protein-ligand complex were found -32.59 kcal/mol. From the results, it was observed that the complex demonstrated thermal equilibrium at 300 K temperature and trajectories showed stable orientation during entire production process.

**3.5.1. The Root Mean Square Deviation (RMSD).** The backbone RMSD was done to observe conformation transpose in protein-ligand complex from the top to bottom orientation of structure. The RMSD graph is represented in Fig 13 which demonstrates that initial deviation of complex over first 5 ns and then showed stability after 7 ns and all over the later stages of simulation. In RMSD, average value of complex was found to be 2.36 Å with an SD of 0.26 Å. Different conformational changes were utilized in order to balance protein-ligand complex.

**3.5.2. Root Mean Square Fluctuation (RMSF).** Protein RMSF: This is useful to observe local variations upon ligand binding with the protein chain. The protein RMSF graph indicates peak area of protein that deviate the most throughout the simulation time as shown in Fig 14. The plot also indicated the considerable fluctuation shown by the N- and C-terminals present on chain end when compared with other region of protein. The reason is that the secondary structure element of protein like α- helix and β-strands are generally firm than the unorganized remaining body of protein and therefore show light deviation as compared to loop areas. The active site residues exhibited less conformational fluctuations in the protein structure.

*Ligand RMSF:* It represented the fluctuation demonstrated by ligand atom in its two-dimensional structure form. The plot provides the thorough information of interaction among ligand fragments with protein during binding event. In Fig 15, the horizontal crooked line exhibits how much ligand fits on protein over simulation period. The protein-ligand complex was initially overlapped on the backbone of protein followed by slight fluctuations

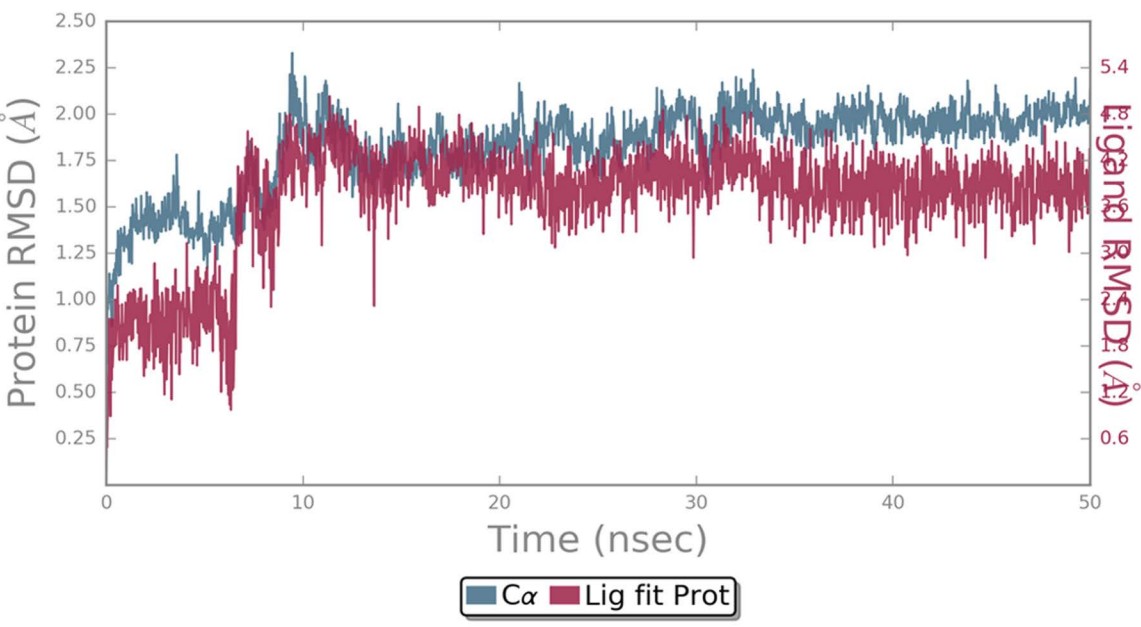

**Fig 13. The RMSD graph of protein-ligand complex.** Simulation was performed using TIP4P water environment with a time interval of 10 ns. (x axis: time scale, y axis: RMSD of protein (left side) and ligand (Left side) in Angstrom respectively.

## Protein RMSF

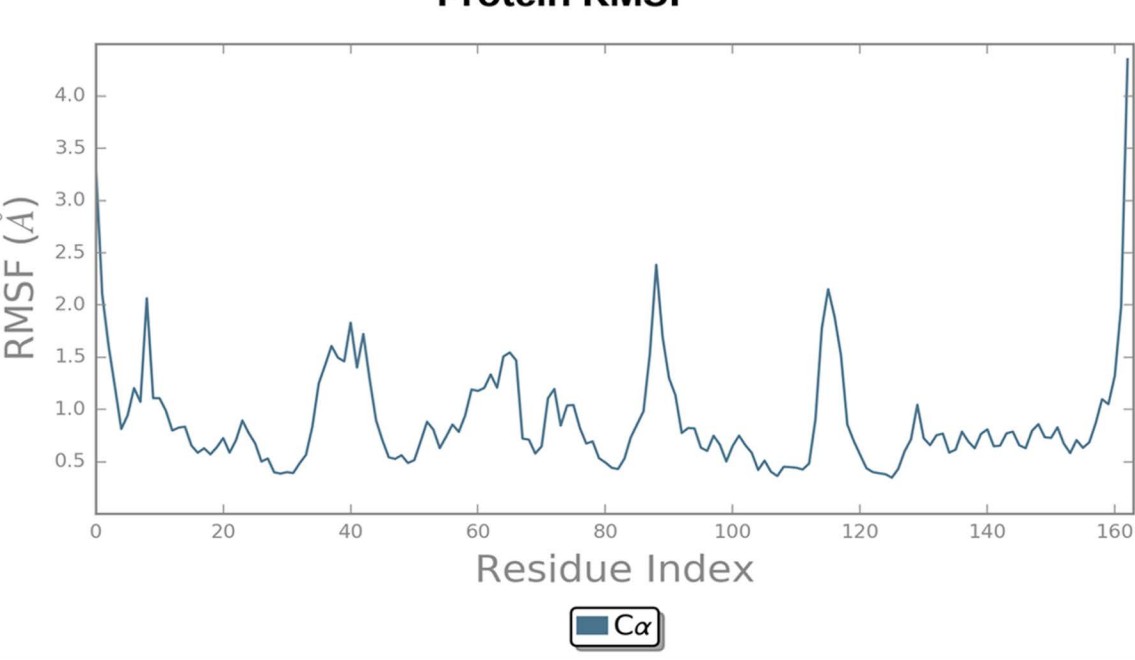

**Fig 14. The RMSF graph of Protein with residue index over the simulation time.**

due to ligand heavy atoms. Thus, it is observed that ligand atoms did not cause any significant change in protein structure upon binding.

### 3.5.3. Hydrogen Bond (H-Bond) analysis.

Hydrogen bonding plays a significant role in drug designing as it has strong relation with drug specificity, metabolism and absorption [88]. Also, it plays an important role in the inhibition of complex molecules because it ensures structural and functional stability. From Fig 16, it is clear that amino acid residue namely TRP60, THR75 and ASP73 showed hydrogen bond interactions with ligand and also this association was found stable throughout the simulation period. Hence, it can be said that protein-ligand complex is relatively stronger during the simulation time. The findings of this in-silico analysis confirmed the structural and functional stability of compounds and their receptor protein complexes, implying that compound D_152 may inhibit *P. aeruginosa* better than other derivatives.

### 3.5.4. Protein ligand Contacts (PLC).

PLC provides information about four types of bonds namely hydrogen bond, hydrophobic interaction, ionic interactions and water bridges. These interactions or contacts are formed between protein and ligand during overall simulation process. Each interaction type has more particular subtypes, which can be discovered using the 'Simulation Interactions Diagram' panel. Fig 17 summarizes the many types of interactions. The four forms of protein-ligand interactions, or "contacts," are water bridges, ionic, hydrogen bonds, and hydrophobic interactions. For instance, a value of 0.7 indicates that the particular contact is maintained 70% of the simulation time. The stacked bar charts are normalized throughout the journey. Values greater than 1.0 could occur because a protein residue could interact with the ligand more than once with the same subtype. Purple slots represent all hydrophobic interactions. In several circumstances, a water molecule acted as a medium for hydrogen-bonded protein-ligand interactions. The hydrogen-bond geometries were thus slightly looser than the usual H-bond definition. On this account, the current geometric criteria for the resulted water-bridges were as follows: a distance of

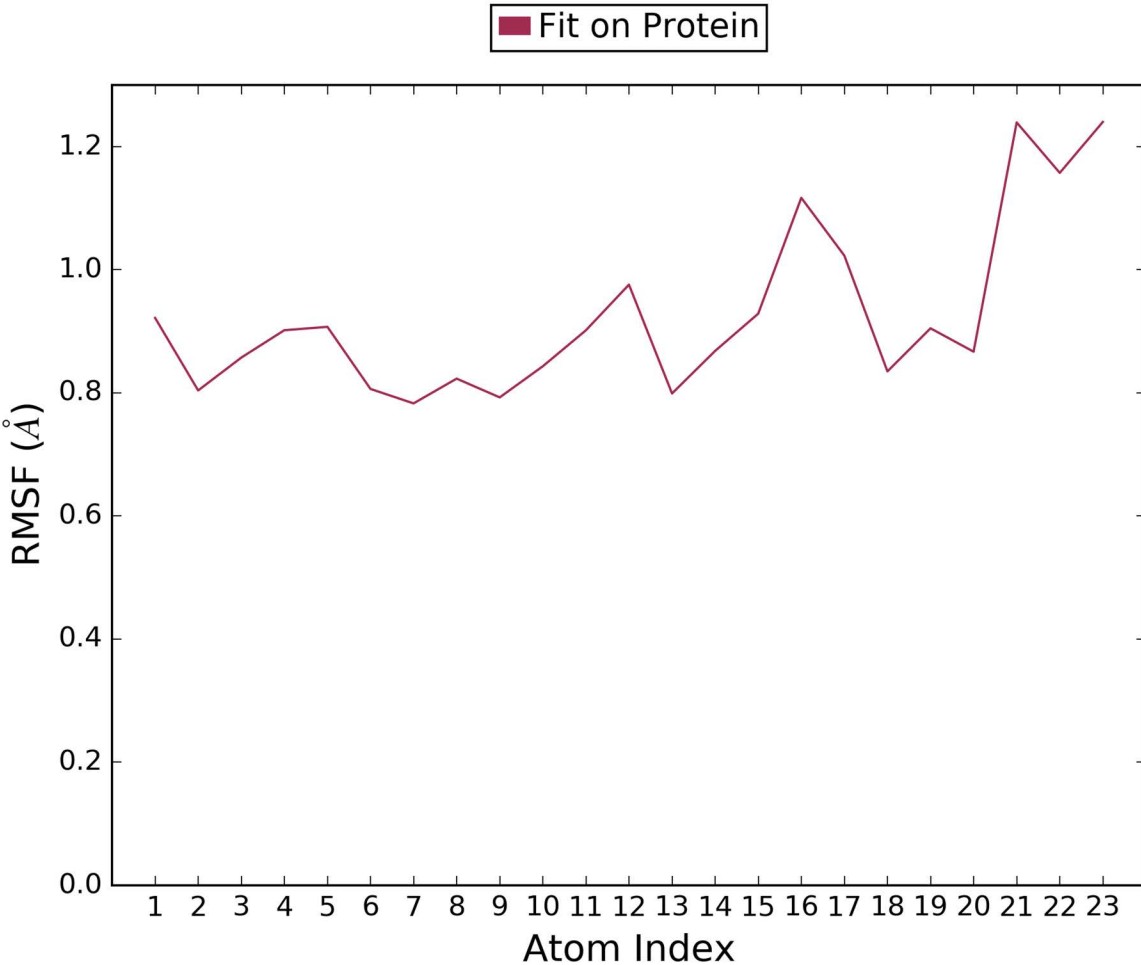

**Fig 15. The RMSF graph of Ligand atoms in correlation with each amino acid residue (mean value over simulation process).**

2.8 Å between the donor and acceptor atoms (D—H···A), a donor angle of ≥110° between the donor-hydrogen-acceptor atoms (D—H···A), and an acceptor angle of ≥90° between the hydrogen-acceptor-bonded atoms (H···A—X). There are three varieties of hydrophobic contacts: π–cation, π–π, and additional non-specific interactions. These interactions typically involve an aromatic or aliphatic group on the ligand and a hydrophobic amino acid, but we have expanded this category to include π–cation interactions as well. At that time, the geometric criteria for hydrophobic interactions were as follows: π–cation for aromatic and charged groups within 4.5 Å, π–π for aromatic groups stacked face-to-face or face-to-edge, and other—associated with a non-specific hydrophobic sidechain within 3.6 Å of the aromatic or aliphatic carbons of a ligand. Two oppositely charged atoms that are within 3.7 Å of one another and do not form a hydrogen bond were evaluated for ionic or polar interactions. PLC analyzed through simulation interaction plot (stacked bar chart) is shown in Fig 17.

### 3.6. DFT calculations

Reactivity is a fundamental idea in chemistry since it is closely related to reaction mechanisms, which makes it possible to comprehend chemical reactions and enhance synthesis

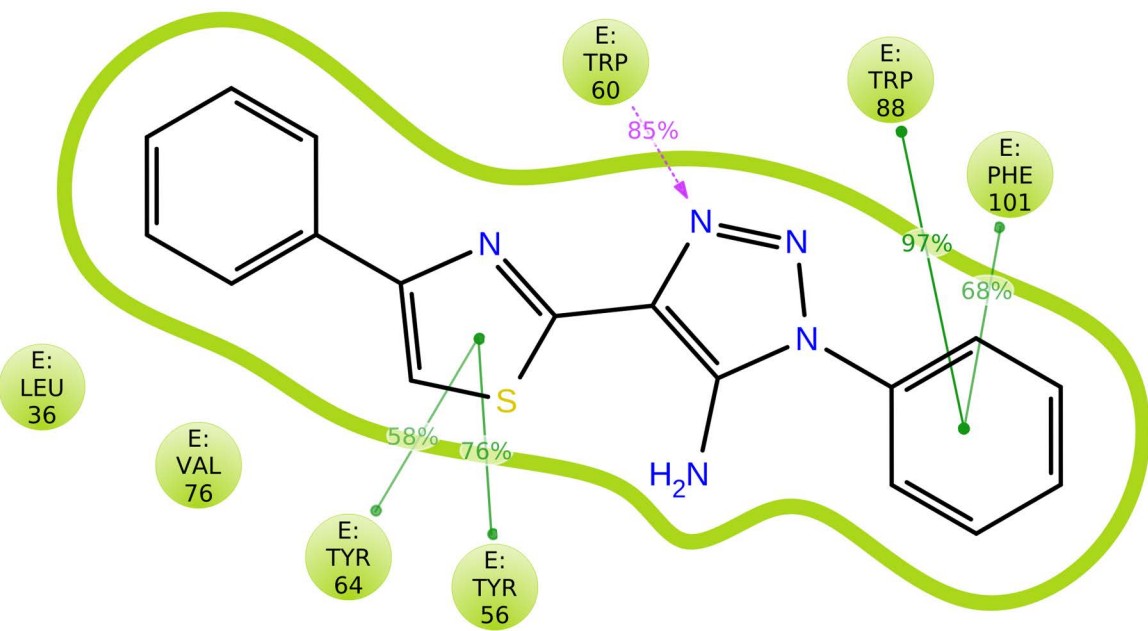

**Fig 16. Schematic representation of ligand atom interactions with amino acid residues presents in protein.** Interactions exhibited more than 30% during simulation period in the selected path track (0.00 through 50.01 nanosec) are highlighted. Purple colored uni-directional arrow represents H-bond interaction with amino acid residual backbone (E:TRP60) of protein. Hydrophobic interactions are shown by amino acid residues with green colored lines.

processes to produce new materials. Confirming the surface catalytic sites, assessing catalytic activity, exposing potential reaction pathways, and creating new compounds with superior catalytic performance are all accomplished by DFT calculations. A thorough analysis of the structure-chemical reactivity relations of thiazole derivatives has been conducted due to their varied biological and industrial significance. The most important structural parameters of title compounds were identified by DFT calculations using the B3LYP functional and 6-311+G(d,p) as basis sets. The influence of the geometric structure on the electronic properties is manifested through the bond lengths and bond angle across the compound backbone. Possessing C1 point group symmetry takes into account the geometry of the chemicals being studied. It was verified that the stationary locations match minima on the Potential Energy Surface by the lack of imaginary frequencies. These calculations did not include solvent adjustments. DFT makes it possible to compute molecule characteristics like energy and optimal geometry [89]. Deterministic functions, such as molecular structures, ionization potentials, electron affinities, and orbital energies, are seen through the utilization of density functional theory (DFT) analysis [90].

*Frontier molecular orbital*: The determination of the reactivity of a chemical moiety is heavily reliant on the characteristics of its frontier orbitals, namely the HOMO and LUMO orbitals. Interactions between the molecule and other species are determined by the frontier molecular orbital. Since it is the outermost orbital with electrons, the HOMO (highest occupied orbital) tends to provide these electrons an electron donor. However, the lowest unoccupied molecular orbital, or LUMO, is assumed to be the innermost orbital with available electron-accepting space. The LUMO energy is therefore directly related to the electron affinity, whereas the HOMO energy is directly related to the ionization potential. Energy gaps, that are crucial for structural stability, are the difference in energy between HOMO and LUMO orbitals. The molecule's kinetic stability and chemical reactivity are described by

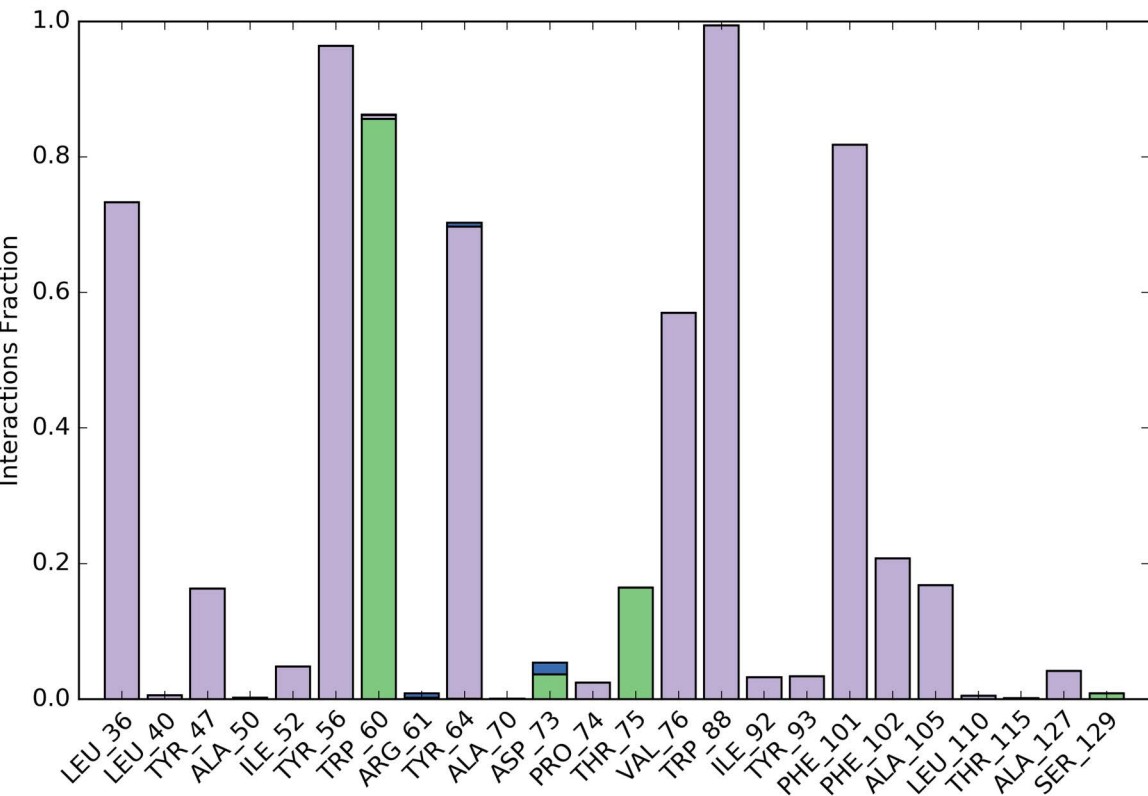

**Fig 17. The stacked bar chart representing protein-ligand contacts of complex.** Green colored slots exhibited hydrogen bond interaction with amino acid residue (TRP60, THR75 and ASP73 with water bridge). All hydrophobic interactions are shown as purple-colored slots.

LUMO. Soft molecules are those having a tiny gap because they are more polarized. Because it is a measure of electron conductivity, the energy gap between HOMO and LUMO has recently been utilized to demonstrate the bioactivity of intramolecular charge transfer (ICT) [91,92]. Molecules having higher values of the Higher Occupied Molecular Orbital (HOMO) are more likely to give electrons to acceptor molecules with lower energy, while molecular orbitals that are vacant tend to take electrons. Molecules with greater HOMO and lower LUMO energy levels exhibit an increased binding potential. The chemical hardness of a system is typically linked to its chemical reactivity and stability. A decrease in value indicates the molecule's capacity to take electrons. The high frontier orbital gap, also known as the HOMO-LUMO gap, denotes the minimum energy at which electron excitation can occur within a molecule. The chemical reactivity of a molecule increases when the HOMO-LUMO gap decreases, leading to less kinetic stability, and vice versa. Both the natural ligand (OHN) and the discovered quantum dot (QSI) (D_152) were subjected to DFT analysis. Compound D_152 and OHN exhibited HOMO-LUMO energy gap values of 4.33 eV and 6.01 eV, respectively. The presence of significant gaps in the molecules suggests the occurrence of active charge transfer, which in turn predicts increased stability and reduced chemical reactivity. Additionally, the bioactivity of a molecule can be attributed to the HOMO-LUMO gap. A molecule with a greater HOMO-LUMO gap is indicative of a less polarized nature, resulting in a more stable form. The DFT analysis demonstrated that both D_152 and OHN exhibit biological activity and possess exceptional stability, as depicted in Fig 18.

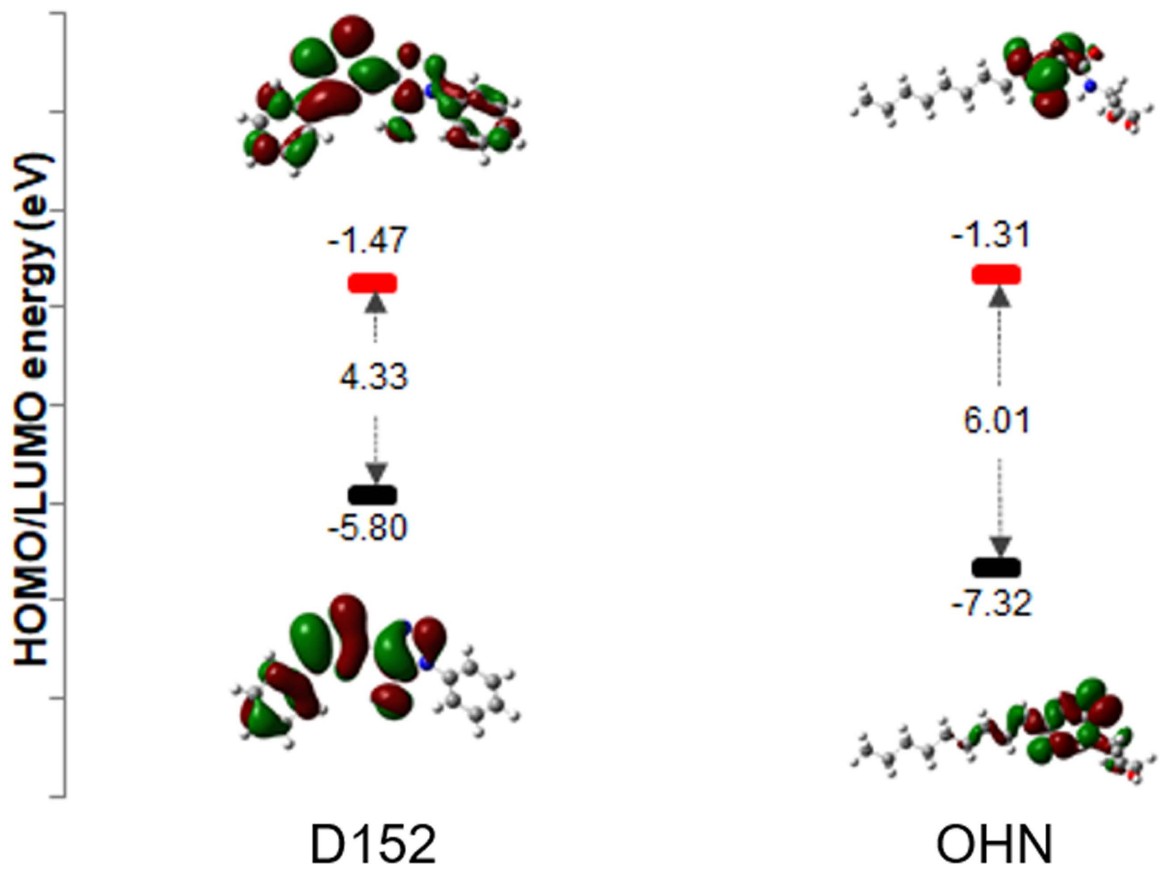

**Fig 18. Model structures of compounds D_152 and OHN (upper) and their corresponding HOMOs and LUMOs (lower).** HOMOs are shown in grey and blue while LUMOs are shown in green and brown.

*Molecular Electrostatic Potential (MEP) Map*: The net electrostatic effect generated at a specific point around the molecule is the total charge distributed (electron + nuclei) and is indicated by the MEP, which is correlated with the molecule's dipole moments, electro negativity, partial charges, and chemical reactivity. It offers a visual way to comprehend the molecule's relative polarity. The proton is drawn to the molecule by its concentrated electron density, which is represented by the negative electrostatic potential. In areas where there is a low electron density and the nuclear charge is not fully insulated, the atomic nuclei repel the proton, which is represented by the positive electrostatic potential. The DFT/B3LYP 6-311 + G(d,p) basis set was used to calculate the MEP for each molecule [91,93]. The molecular electrostatic potential surface (MEPS) is a visual technique employed to determine the position of electron density. The surface exhibits a color scale that corresponds to the charging of electrostatic potential values, with red indicating negativity, green indicating near zero, and blue indicating positivity. The negative potential of MEP associated with reactive electrophilic sites is indicated by the red color, whereas the positive potential zone reflects the appropriate centers of the nucleophilic attack, as indicated by the blue color [94]. A surface map of an electron density and electrostatic potential shows the molecules' size, shape, charge density, and reactive sites. Red denotes the areas with the highest negative electrostatic potential, blue the areas with the highest positive electrostatic potential, and green the area with zero potential. These colors correspond to the various electrostatic potential levels. The MEP surfaces for compound D_152 and OHN are depicted in Fig 19. The magnetic resonance spectroscopy

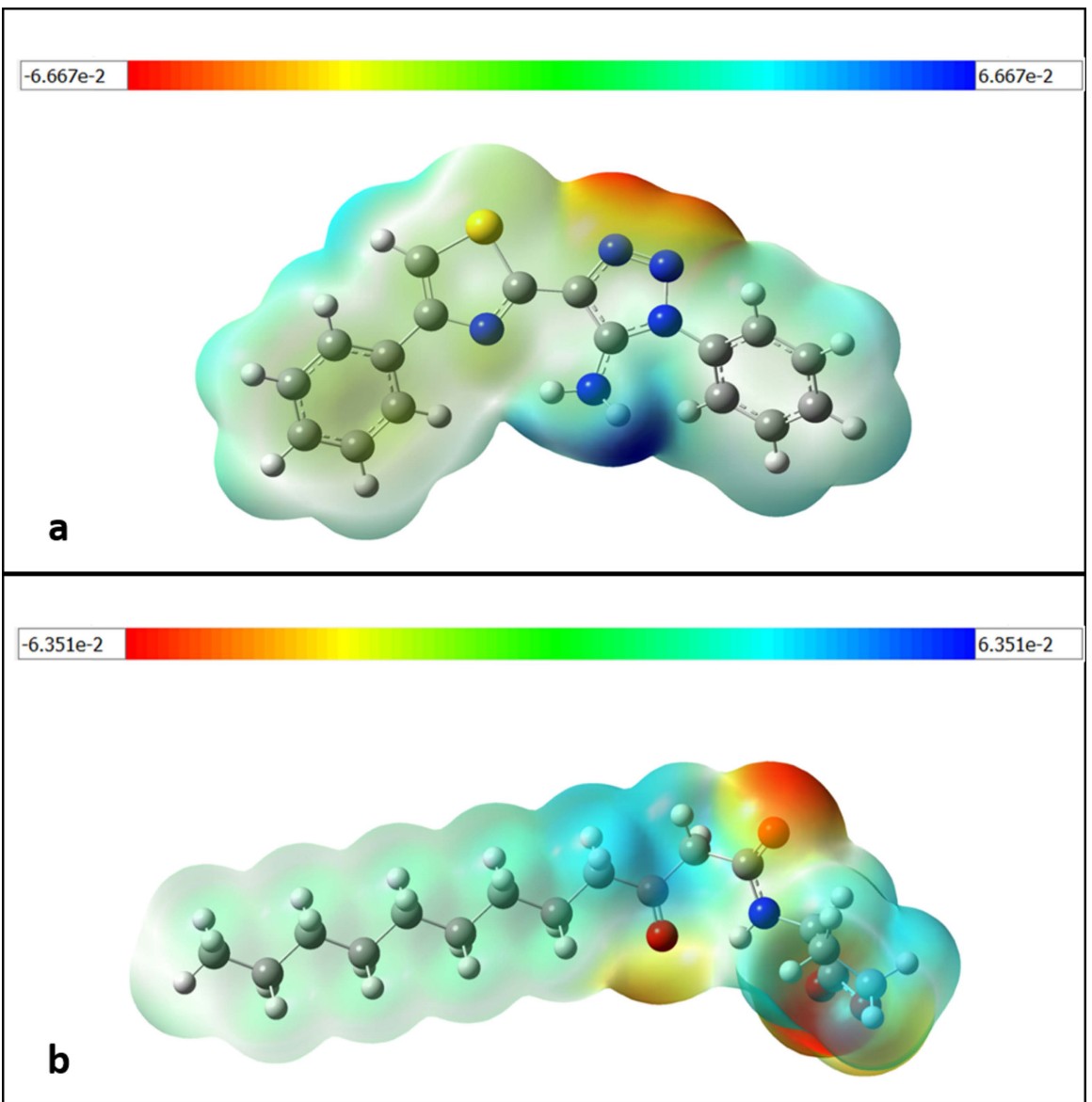

**Fig 19. Molecular electrostatic potential (MEP) maps (Kcal/mol) drawn for a selected iso-surface value for (a) compound D_152; (b) OHN in ground state calculated using DFT calculations.** Red color with negative value represents minimum electrostatic potential (bound loosely or excess electrons) available for electrophilic attack. Blue region indicates the positive value for maximum electrostatic potential and acts opposite.

(MEP) map revealed distinct zones for compound D_152 and OHN, with values ranging from -6.667 to +6.667 and -6.351 to +6.351 kcal/mol, respectively. Both compounds displayed the most significant negative values in the amide group and O atom of the furan ring. Both compounds displayed double unsaturation of the nitrogen atom, and the inclusion of an oxygen atom next to the carbonyl group in the carbonic chain, together with large substitutions, suggests a greater electron density around the furan ring. In addition, the electron density is concentrated on the nitrogen (N) and oxygen (O) atoms inside the amide group, namely within the red and yellow regions. The chemical D_152's ability to detect quorum can be related to the electron density surrounding the amide group, O atom of the furan ring, and other atoms. This observation implies that the interaction between the compound's highly active important

structural areas, specifically those that are emphasized, and the active site of the protein (LasR) is involved in a biological recognition mechanism.

## 4. Conclusion

The reference molecule (TC), which is a recognized QSI of *P. aeruginosa*, was subjected to molecular docking simulations utilizing the intuitive interface of the molecular docking application AutoDock Vina. For the LasR and CviR receptors, molecular coupling simulations based on crystal structure were performed. The docking simulations conducted with Pymol software to generate the molecular complexes of LasR-OHN were superimposed onto the co-crystallized structure of LasR. The hydrophobic and hydrogen bonding interactions of these complexes with amino acid residues were identical to those of the native ligand (OHN) in its co-crystallized state. Similar to the native ligand (HLC) in its co-crystallized state, the molecular docking simulation utilizing the co-crystallized structure of CviR (PDB ID: 3QP5) uncovered hydrophobic and hydrogen bond interactions with the amino acid. Using TC as the template compound, ligand/structure-based virtual screening was performed; CviR received a docking score of -9.427, while LasR received a score of -12.4. By utilizing the ZINC database, structural similarity was to be identified. An exhaustive collection of 800 compounds was acquired from the ZINC database and subsequently underwent molecular docking-based virtual screening utilizing Autodock Vina. A total of 29 substances were assessed during the selection process, with consideration given to their pharmacokinetic, toxicological, and physicochemical characteristics. On the basis of combined docking and ADME/T results, the compounds with ZINC IDs ZINC16650743 (D_152), ZINC67334656 (D_153), and ZINC32760540 (L_331) were identified as QSIs with putative antagonistic activity against the LasR protein of *P. aeruginosa*. The data that was obtained revealed the compounds' structural similarity, compatibility with the LasR protein, and safer ADMET capabilities. The three compounds that were identified (D_152, D_153, and L_331) underwent an in-silico molecular docking analysis utilizing the CviR receptor. Furthermore, molecular dynamic simulation analyses were performed on compound D_152 to detect any atomic motion within the molecules over a predetermined period of time. Molecular dynamics simulations revealed that the protein-ligand complex of D_152 demonstrated significant binding, except for slight conformational alterations detected in the tail regions (specifically, the N and C-terminal sections) of the protein structure. D_152 underwent a Density Functional Theory (DFT) analysis in comparison to OHN. The analysis using density functional theory (DFT) revealed that the molecular structure of both compounds was exceptionally stable, and their quorum sensing activity was primarily attributed to a specific structural region. The feasibility of employing a thiazole derivative as an innovative quorum sensing inhibitor (LasR inhibitor) was illustrated through *in-silico* analyses.

The outcomes of this study underscore the potency of thiazole derivatives as novel QSIs targeting the LasR system, thereby providing an alternative to lactone-based QS inhibitors. Indeed, this study is primarily theoretical; as much as it provides rather strong predictive ideas, it is, however, critical, and limitations exist on the true replicable nature of biological complexities via an *in silico* approach. Therefore, experimental validations hold a giant share in proving the biological efficacy and safety of drug molecules in the real-life biological world. Parenting the molecular dynamics with the simulation period lasting more than 100 nanoseconds and some added compounds would provide quite a considerable validation of these results in future.

Subsequent studies should mainly focus on broad-spectrum biological validation to assess the anticancer potential of identified compounds. These require further clinical trials to determine effective treatment strategies against the *LasR* quorum-sensing system, and possibly lead to innovative interventions to combat multiresistant *P. aeruginosa*. The groundwork provided

by this study could be used as the foundation for the design of new anti-virulence therapies which represent a radical change in the approach towards the management of bacterial infections.

## Author contributions

**Conceptualization:** Snigdha Bhardwaj, Kandasamy Nagarajan.

**Data curation:** Halima Mustafa Elagib, Sadaf Anwar.

**Funding acquisition:** Halima Mustafa Elagib, Mohd Adnan Kausar.

**Investigation:** Tulika Bhardwaj.

**Methodology:** Tulika Bhardwaj.

**Supervision:** Mohd Adnan Kausar.

**Writing – original draft:** Snigdha Bhardwaj, Kandasamy Nagarajan.

**Writing – review & editing:** Mohammad Zeeshan Najm, Mohd Adnan Kausar.

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
