## [Decision Letter · Decision Letter 0]

23 Oct 2024

PONE-D-24-31938Molecular simulation-based investigation of thiazole derivatives as potential quorum sensing inhibitors of Pseudomonas aeruginosaPLOS ONE

Dear Dr. Kausar,

Thank you for submitting your manuscript to PLOS ONE. After careful consideration, we feel that it has merit but does not fully meet PLOS ONE’s publication criteria as it currently stands. Therefore, we invite you to submit a revised version of the manuscript that addresses the points raised during the review process.

Your manuscript requires substantial revisions, particularly in terms of clarifying the study's focus, addressing methodological limitations, and including missing data. Please carefully address all of the reviewers' comments.

We look forward to receiving your revised manuscript.

Kind regards,

Tung Truong-Thanh

Academic Editor

PLOS ONE

“This research has been funded by Deputy for Research & Innovation, Ministry of Education through Initiative of Institutional Funding at University of Ha’il – Saudi Arabia through project number IFP-22 103.”

“This research has been funded by Deputy for Research & Innovation, Ministry of Education through Initiative of Institutional Funding at University of Ha’il – Saudi Arabia through project number IFP-22 103.”

“This research has been funded by Deputy for Research & Innovation, Ministry of Education through Initiative of Institutional Funding at University of Ha’il – Saudi Arabia through project number IFP-22 103.”

5. Please include your tables as part of your main manuscript and remove the individual files. Please note that supplementary tables (should remain/ be uploaded) as separate "supporting information" files

Additional Editor Comments:

A group of expert reviewers and I have assessed your submission and feel that substantial revisions must be done before we can consider it further. Please see the attached reviewer comments for further details about necessary revisions.

Reviewers' comments:

Reviewer's Responses to Questions

**Comments to the Author**

1. Is the manuscript technically sound, and do the data support the conclusions?

Reviewer #1: Partly

Reviewer #2: Yes

2. Has the statistical analysis been performed appropriately and rigorously? 

Reviewer #1: No

Reviewer #2: Yes

3. Have the authors made all data underlying the findings in their manuscript fully available?

Reviewer #1: No

Reviewer #2: Yes

4. Is the manuscript presented in an intelligible fashion and written in standard English?

Reviewer #1: Yes

Reviewer #2: Yes

5. Review Comments to the Author

Reviewer #1: Snigdha Bhardwaj and colleagues reported an in-silico screening of possible inhibitors of the Quorum Sensing system in P. aeruginosa. The study described different approaches to eventually identify 29 compounds that were better investigated for their pharmacokinetics and toxicity in silico.

Actually, the study focuses on thiazole derivatives and LasR interaction. Therefore, the Authors need to better clarify in their study why they decided to take into consideration only thiazole derivatives (actually, several different compounds have been reported in QS inhibition in P. aeruginosa) and why they focused on LasR system. Given the receptor specificity of the study, I suggest introducing “LasR” in the title of the Manuscript (something like “thiazole derivatives as potential LasR inhibitors of Pseudomonas aeruginosa”)

The study is rich in different approaches; however, the Authors did not mention about the limitations of the study and did not describe the methods used in data interpretation. Moreover, as the results were obtained in silico, the meanings of the study need to be re-considered, and the last sentence (lines 457-459) needs to be carefully reconsidered.

Several others major points need attention:

1- In the Introduction (line 73), the Authors wrote that P. aeruginosa possesses two unique QS circuits. Actually four QS pathways were identified in P. aeruginosa, namely Las, Rhl, Pqs, and Iqs (DOI: 10.1126/science.aaw1629). Moreover, in P. aeruginosa, QS controls not only biofilm formation but also and rhamnolipids, pyocyanin, and pyoveridin production. Authors should upload the information.

2- The Authors should better describe the advantages of studying the docking in both CviR and LasR: what do these data mean for the present study as they gave different docking scores even against the TC?

3- Pharmacokinetic and Toxicity (ADME/T) Studies should also include studies about the stability of the potential inhibitors. Considering the administration route, are the compound stable in biological fluids?

4- —Line 300: The authors should detail why they selected the value of -12.0 kcal/mol. Maybe changing the reference compound TC would change the cut-off values, too?

5- Line 308: The main text does not report Table 1. The Reviewer cannot review the table. Table should report the docking score values for the 800 test compounds (maybe grouped by range values) plus more details (molecular structures) about the 29 selected molecules. Line 324: at the same, Table 2 is not reported in the main text and the Reviewer was not able to review the Table 2. Line 333: the same for Table 3

6- The results of the Molecular Dynamics Simulations (para 3.5) are not described; it is not clear which molecules are involved in the complex formation. The Authors should justify the parameters set for the experiments (10 ns).

7- Line 372: To improve comprehension, Authors should better describe what they mean as “ligand”

8- No description of the results has been provided for Protein-ligand Contacts (para 3.5.4).

Minor points

Line 229: name of bacteria in italic

Line 230. What is C.?

Line 320: ouome?

Line 342: closing bracket is missing

Line 371: please correct the mistake

Line 418: please, better formulate the concept “The chemical D_152's ability to detect quorum” as it is not clear

Line 425, 443, and 457: name of bacteria in italic

Figures 1, 2, 10, 11, 12 are not mentioned in the main text

Reviewer #2: Editorial Office

PLOS ONE

Thank you very much for choosing me as a potential reviewer for the Manuscript Title: Molecular simulation-based investigation of thiazole derivatives as potential 1 quorum sensing inhibitors of Pseudomonas aeruginosa.

The manuscript needs minor revision. Detailed explanation of some general and specific points should be given:

Comments

-The manuscript needs careful proofreading to correct some punctuation and spelling errors.

-While the manuscript is generally well-organized and clear, breaking down some long sentences could enhance readability.

-Minor corrections are needed in the abstract. Emphasize the aim, novelty, and outcomes of the work.

-The introduction requires enhancement. Please provide more insight into the rationale behind selecting the specific compounds for your study.

-Highlight the advantages of Molecular docking calculations in the relevant sections, Refer to and cite the following related papers that utilized advantages of Molecular docking calculations to enhance the Molecular docking part: (https://doi.org/10.1021/acsomega.3c03767) (https://doi.org/10.1016/j.jobb.2023.09.001) (https://doi.org/10.1002/jhet.4752).

- It is important to give the reason for whay selecte the used PDB ID protein in this study

-Clarify the preparation of the protein PDB file for docking. Were additional ligands or water molecules removed? Were hydrogen atoms and charges added? Was the 3D optimized output used for docking the compounds' structures? Explain this in the relevant section. Refer to and cite the following papers to enhance the docking part: (https://doi.org/10.1038/s41598-023-46602-1) (https://doi.org/10.4314/bcse.v38i1.12) (https://doi.org/10.1002/aoc.7299).

-Include the IUPAC names for the compounds.

-The conclusions are generally well-supported by the data and analyses presented.

-Highlight the advantages of DFT calculations and the HOMO-LUMO, Hardness, MEP calculations in the relevant section, and cite the following papers that utilized DFT calculations. Refer to and cite the following articles to enhance the DFT part: (https://doi.org/10.1016/j.molstruc.2023.136264) (https://doi.org/10.1002/aoc.7262) (https://doi.org/10.1016/j.inoche.2023.111486).

- What is about the future recommendations!

- Names of all bacterial strains should written as itallic

- What about Molecular docking validation!

6. PLOS authors have the option to publish the peer review history of their article (what does this mean? ). If published, this will include your full peer review and any attached files.

**Do you want your identity to be public for this peer review?** For information about this choice, including consent withdrawal, please see our Privacy Policy .

Reviewer #1: **Yes**

Reviewer #2: No

---

## [Author Response · Author response to Decision Letter 0]

29 Nov 2024

Review Comments to the Author

Reviewer #1: Snigdha Bhardwaj and colleagues reported an in-silico screening of possible inhibitors of the Quorum Sensing system in P. aeruginosa. The study described different approaches to eventually identify 29 compounds that were better investigated for their pharmacokinetics and toxicity in silico.

Actually, the study focuses on thiazole derivatives and LasR interaction. Therefore, the Authors need to better clarify in their study why they decided to take into consideration only thiazole derivatives (actually, several different compounds have been reported in QS inhibition in P. aeruginosa) and why they focused on LasR system. Given the receptor specificity of the study, I suggest introducing “LasR” in the title of the Manuscript (something like “thiazole derivatives as potential LasR inhibitors of Pseudomonas aeruginosa”)

Title has been changed to “Molecular simulation-based investigation of thiazole derivatives as potential LasR inhibitors of Pseudomonas aeruginosa”

The study is rich in different approaches; however, the Authors did not mention about the limitations of the study and did not describe the methods used in data interpretation. Moreover, as the results were obtained in silico, the meanings of the study need to be re-considered, and the last sentence (lines 457-459) needs to be carefully reconsidered.

The last statement has been revised with clear explanation (Page no. 26, red coloured text).

Several others major points need attention:

1. In the Introduction (line 73), the Authors wrote that P. aeruginosa possesses two unique QS circuits. Actually, four QS pathways were identified in P. aeruginosa, namely Las, Rhl, Pqs, and Iqs (DOI: 10.1126/science.aaw1629).Moreover, in P. aeruginosa, QS controls not only biofilm formation but also and rhamnolipids, pyocyanin, and pyoveridin production. Authors should upload the information.

The detail for QS circuit/pathways that P. aeruginosa possesses (reference 11,12- on page 4-5) has been added along with the details of different virulent factors (reference 18,19) on page 5-6.

References:

Miranda SW, Asfahl KL, Dandekar AA, Greenberg EP. Pseudomonas aeruginosa Quorum Sensing. Adv Exp Med Biol. 2022;1386:95-115. doi: 10.1007/978-3-031-08491-1_4. PMID: 36258070; PMCID: PMC9942581.

Welsh MA, Blackwell HE. Chemical Genetics Reveals Environment-Specific Roles for Quorum Sensing Circuits in Pseudomonas aeruginosa. Cell Chem Biol. 2016 Mar 17;23(3):361-9. doi: 10.1016/j.chembiol.2016.01.006. Epub 2016 Feb 18. PMID: 26905657; PMCID: PMC4798878.

Moura-Alves P, Puyskens A, Stinn A, Klemm M, Guhlich-Bornhof U, Dorhoi A, Furkert J, Kreuchwig A, Protze J, Lozza L, Pei G, Saikali P, Perdomo C, Mollenkopf HJ, Hurwitz R, Kirschhoefer F, Brenner-Weiss G, Weiner J 3rd, Oschkinat H, Kolbe M, Krause G, Kaufmann SHE. Host monitoring of quorum sensing during Pseudomonas aeruginosa infection. Science. 2019 Dec 20;366(6472):eaaw1629. doi: 10.1126/science.aaw1629. PMID: 31857448.

Raman Pachaiappan, Tharun Prasanna Rajamuthu, Ananya Sarkar, Pradiksha Natrajan, Nagasathiya Krishnan, Meenakumari Sakthivelu, Palaniyandi Velusamy, Palaniappan Ramasamy, Subash C.B. Gopinath, N-acyl-homoserine lactone mediated virulence factor(s) of Pseudomonas aeruginosa inhibited by flavonoids and isoflavonoids, Process Biochemistry, Volume 116, 2022, Pages 84-93, ISSN 1359-5113, https://doi.org/10.1016/j.procbio.2022.02.024.

2. The Authors should better describe the advantages of studying the docking in both CviR and LasR: what do these data mean for the present study as they gave different docking scores even against the TC?

The advantages of studying the docking in both CviR and LasR has been added under point 3.1 (reference 68) on page 12-13.

Reference

Baloyi IT, Adeosun IJ, Yusuf AA, Cosa S. In Silico and In Vitro Screening of Antipathogenic Properties of Melianthus comosus (Vahl) against Pseudomonas aeruginosa. Antibiotics (Basel). 2021 Jun 5;10(6):679. doi: 10.3390/antibiotics10060679. PMID: 34198845; PMCID: PMC8230066.

3. Pharmacokinetic and Toxicity (ADME/T) Studies should also include studies about the stability of the potential inhibitors. Considering the administration route, are the compound stable in biological fluids?

The description about the stability aspects of the potential inhibitors through ADMET studies have been added under point 3.3 (reference 80,81,82) on page 17.

References

George Lambrinidis, Theodosia Vallianatou, Anna Tsantili-Kakoulidou, In vitro, in silico and integrated strategies for the estimation of plasma protein binding. A review, Advanced Drug Delivery Reviews, Volume 86, 2015, Pages 27-45, ISSN 0169-409X, https://doi.org/10.1016/j.addr.2015.03.011.

Nisha CM, Kumar A, Nair P, Gupta N, Silakari C, Tripathi T, Kumar A. Molecular Docking and In Silico ADMET Study Reveals Acylguanidine 7a as a Potential Inhibitor of β-Secretase. Adv Bioinformatics. 2016;2016:9258578. doi: 10.1155/2016/9258578. Epub 2016 Apr 10. PMID: 27190510; PMCID: PMC4842033.

Zakari Ya'u Ibrahim, Adamu Uzairu, Gideon Shallangwa, Stephen Abechi, Molecular docking studies, drug-likeness and in-silico ADMET prediction of some novel β-Amino alcohol grafted 1,4,5-trisubstituted 1,2,3-triazoles derivatives as elevators of p53 protein levels, Scientific African, Volume 10, 2020, e00570, ISSN 2468-2276, https://doi.org/10.1016/j.sciaf.2020.e00570.

4. Line 300: The authors should detail why they selected the value of -12.0 kcal/mol. Maybe changing the reference compound TC would change the cut-off values, too?

The justification of selecting cut off value is added under point 3.2 (reference 79) on page 16.

Reference

Khidre, R.E., Radini, I.A.M. Design, synthesis and docking studies of novel thiazole derivatives incorporating pyridine moiety and assessment as antimicrobial agents. Sci Rep 11, 7846 (2021). https://doi.org/10.1038/s41598-021-86424-7

5. Line 308: The main text does not report Table 1. The Reviewer cannot review the table. Table should report the docking score values for the 800 test compounds (maybe grouped by range values) plus more details (molecular structures) about the 29 selected molecules. Line 324: at the same, Table 2 is not reported in the main text and the Reviewer was not able to review the Table 2. Line 333: the same for Table 3

All the Tables 1,2,3 provided as a separate document. Please see “other” file.

Secondly, providing Table with the docking score values for all 800 test compounds (even with grouped range values) in the manuscript will not put much impact as we have justified in the manuscript for selecting compounds with cut off value above or equal -12 kcal/mol to extract the potential compounds having good binding affinity with LasR receptor. So employing those shortlisted hits in the study.

6. The results of the Molecular Dynamics Simulations (para 3.5) are not described; it is not clear which molecules are involved in the complex formation. The Authors should justify the parameters set for the experiments (10 ns).

The justification of protein -ligand complex formation along with parameter details is added under point 3.5 (reference 84) on page 18.

Reference

Hollingsworth SA, Dror RO. Molecular Dynamics Simulation for All. Neuron. 2018 Sep 19;99(6):1129-1143. doi: 10.1016/j.neuron.2018.08.011. PMID: 30236283; PMCID: PMC6209097.

7. Line 372: To improve comprehension, Authors should better describe what they mean as “ligand”.

Explain in description under point 3.5 (reference 84) on page 18.

8. No description of the results has been provided for Protein-ligand Contacts (para 3.5.4).

The description for Protein-ligand Contacts have been added under point 3.5.4 on page 20,21.

Minor points- Revised as coloured red

1. Line 229: name of bacteria in italic- DONE

2. Line 230. What is C.? - DONE

3. Line 320: ouome? - DONE

4. Line 342: closing bracket is missing - DONE

5. Line 371: please correct the mistake - DONE

6. Line 418: please, better formulate the concept “The chemical D_152's ability to detect quorum” as it is not clear - DONE

7. Line 425, 443, and 457: name of bacteria in italic - DONE

8. Figures 1, 2, 10, 11, 12 are not mentioned in the main text - DONE

Reviewer #2: Editorial Office

The manuscript needs minor revision. Detailed explanation of some general and specific points should be given:

Comments

1. The manuscript needs careful proofreading to correct some punctuation and spelling errors.

Authors agree with the reviewer’s suggestion and justified modifications are made in the manuscript.

2. While the manuscript is generally well-organized and clear, breaking down some long sentences could enhance readability.

Revised wherever applicable all over the manuscript.

3. Minor corrections are needed in the abstract. Emphasize the aim, novelty, and outcomes of the work.

Abstract revised with special emphasis on aim, novelty and outcome (Page 2-3).

4. The introduction requires enhancement. Please provide more insight into the rationale behind selecting the specific compounds for your study.

The insight into the rationale behind selecting the specific compounds for this study has been added (reference 33-37) on page 7.

References

R Santosh, MK Selvam, SU Kanekar, GK Nagaraja, M Kumar. Design, Synthesis, DNA Binding, and Docking Studies of Thiazoles and Thiazole‐Containing Triazoles as Antibacterials ChemistrySelect 3 (14), 3892-3898. https://doi.org/10.1002/slct.201800222

Kartsev V, Geronikaki A, Zubenko A, Petrou A, Ivanov M, Glamočlija J, Sokovic M, Divaeva L, Morkovnik A, Klimenko A. Synthesis and Antimicrobial Activity of New Heteroaryl(aryl) Thiazole Derivatives Molecular Docking Studies. Antibiotics. 2022; 11(10):1337. https://doi.org/10.3390/antibiotics11101337

Ratrey P, Das Mahapatra A, Pandit S, Hadianawala M, Majhi S, Mishra A, Datta B. Emergent antibacterial activity of N-(thiazol-2-yl)benzenesulfonamides in conjunction with cell-penetrating octaarginine. RSC Adv. 2021 Aug 25;11(46):28581-28592. doi: 10.1039/d1ra03882f. PMID: 35478531; PMCID: PMC9038147.

Abdelsamie AS, Hamed MM, Schütz C, Röhrig T, Kany AM, Schmelz S, Blankenfeldt W, Hirsch AKH, Hartmann RW, Empting M. Discovery and optimization of thiazole-based quorum sensing inhibitors as potent blockers of Pseudomonas aeruginosa pathogenicity. Eur J Med Chem. 2024 Oct 5;276:116685. doi: 10.1016/j.ejmech.2024.116685. Epub 2024 Jul 16. PMID: 39042991.

More, P.G., Karale, N.N., Lawand, A.S. et al. Synthesis and anti-biofilm activity of thiazole Schiff bases. Med Chem Res 23, 790–799 (2014). https://doi.org/10.1007/s00044-013-0672-7

5. Highlight the advantages of Molecular docking calculations in the relevant sections, Refer to and cite the following related papers that utilized advantages of Molecular docking calculations to enhance the Molecular docking part:

The advantages of Molecular docking calculations under introduction part has been added (reference 24-26) on page 6.

References

Ahmed M. El-Saghier, Aly Abdou, Mounir A. A. Mohamed, Hany M. Abd El-Lateef, and Asmaa M. Kadry, Novel 2-Acetamido-2-ylidene-4-imidazole Derivatives (El-Saghier Reaction): Green Synthesis, Biological Assessment, and Molecular Docking. ACS Omega 2023 8 (33), 30519-30531 DOI: 10.1021/acsomega.3c03767

Ranjan K. Mohapatra, Ahmed Mahal, Azaj Ansari, Manjeet Kumar, Jyoti Prakash Guru, Ashish K. Sarangi, Aly Abdou, Snehasish Mishra, Mohammed Aljeldah, Bashayer M. AlShehail, Mohammed Alissa, Mohammed Garout, Ahmed Alsayyah, Ahmad A. Alshehri, Ahmed Saif, Abdulaziz Alqahtani, Fahd A. Alshehri, Aref A. Alamri, Ali A. Rabaan, Comparison of the binding energies of approved mpox drugs and phytochemicals through molecular docking, molecular dynamics simulation, and ADMET studies: An in silico approach, Journal of Biosafety and Biosecurity, Volume 5, Issue 3, 2023, Pages 118-132, ISSN 2588-9338, https://doi.org/10.1016/j.jobb.2023.09.001.

Ahmed M. El‐Saghier, Souhaila S. Enaili, Aly Abdou, Amany M. Hamed, Asmaa M. Kadry, An operationally simple, one‐pot, convenient synthesis, and in vitro anti‐inflammatory activity of some new spirotriazolotriazine derivatives, Journal of Heterocyclic Chemistry, Volume 61, Issue 1, 2024, Pages 146-162, ISSN 0022-152X, https://doi.org/10.1002/jhet.4752.

6. It is important to give the reason for why selected the used PDB ID protein in this study

The justification for the selection of used PDB ID protein in this study has been added under 2.2 heading “Virtual Screening based on Structure Similarity” (reference 54) on page 10.

Reference

Bottomley MJ, Muraglia E, Bazzo R, Carfì A. Molecular insights into quorum sensing in the human pathogen Pseudomonas aeruginosa from the structure of the virulence regulator LasR bound to its autoinducer. J Biol Chem. 2007 May 4;282(18):13592-600. doi: 10.1074/jbc.M700556200. Epub 2007 Mar 15. PMID: 17363368.

7. Clarify the preparation of the protein PDB file for docking. Were additional ligands or water molecules removed? Were hydrogen atoms and charges added? Was the 3D optimized output used for docking the compounds ‘structures? Explain this in the relevant section. Refer to and cite the following papers to enhance the docking part:(

The process of preparation of the protein PDB file for docking has been added under The below mentioned suggested references are added as 45,46,47 in the manuscript, under 2.1.3. Protein Preparation on pages 8-9.

References

El-Saghier, A.M., Enaili, S.S., Kadry, A.M. et al. Green synthesis, biological and molecular docking of some novel sulfonamide thiadiazole derivatives as potential insecticidal against Spodoptera littoralis. Sci Rep 13, 19142 (2023). https://doi.org/10.1038/s41598-023-46602-1

Hany M. Abd El-Lateef, Ali M. Ali, Mai M. Khalaf, Aly Abdou. New iron(III), cobalt(II), nickel(II), copper(II), zinc(II) mixed-ligand complexes: Synthesis, structural, DFT, molecular docking and antimicrobial analysis, Bull. Chem. Soc. Ethiop. 2024, 38(1), 147-166. https://dx.doi.org/10.4314/bcse.v38i1.12

Hany M. Abd El-Lateef, Mai M. Khalaf, Amer A. Amer, Antar A. Abdelhamid, Aly Abdou. Antibacterial, antifungal, anti-inflammatory evaluation, molecular docking, and density functional theory exploration of 2-(1H-benzimidazol-2-yl)guanidine mixed-ligand complexes: Synthesis and characterization. Appl. Organomet. Chem, Volume38, Issue1 ,January 2024, e7299 https://doi.org/10.1002/aoc.7299

8. Include the IUPAC names for the compounds.

The IUPAC names are added in the table no. 1 for shortlisted derivatives.

9. The conclusions are generally well-supported by the data and analyses presented.

NA

10. Highlight the advantages of DFT calculations and the HOMO-LUMO, Hardness, MEP calculations in the relevant section, and cite the following papers that utilized DFT calculations. Refer to and cite the following articles to enhance the DFT part:

Advantages of DFT calculations and the HOMO-LUMO, Hardness, MEP calculations in the relevant section has been added as red coloured text. The below mentioned suggested references [as 90,92,93,94 in the manuscript, pages from 21-24] have been added to support the information.

References

Aly Abdou, Omran A. Omran, Jabir H. Al-Fahemi, Rabab S. Jassas, Munirah M. Al-Rooqi, Essam M. Hussein, Ziad Moussa, Saleh A. Ahmed, Lower rim thiacalixarenes derivatives incorporating multiple coordinating carbonyl groups: Synthesis, characterization, ion-responsive ability and DFT computational analysis, Journal of Molecular Structure, Volume 1293, 2023, 136264, ISSN 0022-2860, https://doi.org/10.1016/j.molstruc.2023.136264.

Bendjeddou, A & Abbaz, T & Gouasmia, Abdelkrim & Villemin, Didier. (2016). Molecular Structure, HOMO-LUMO, MEP and Fukui Function Analysis of Some TTF-donor Substituted Molecules Using DFT (B3LYP) Calculations. International Research Journal of Pure and Applied Chemistry. 12. 1-9. 10.9734/IRJPAC/2016/27066.

El-Lateef, Hany & Khalaf, Mai & Heakal, Fakiha & Abdou, Aly. (2023). Fe(III), Ni(II), and Cu(II

---

## [Decision Letter · Decision Letter 1]

27 Dec 2024

PONE-D-24-31938R1Molecular simulation-based investigation of thiazole derivatives as potential LasR inhibitors of Pseudomonas aeruginosaPLOS ONE

Dear Dr. Kausar,

Thank you for submitting your manuscript to PLOS ONE. After careful consideration, we feel that it has merit but does not fully meet PLOS ONE’s publication criteria as it currently stands. Therefore, we invite you to submit a revised version of the manuscript that addresses the points raised during the review process.  Please submit your revised manuscript by Feb 10 2025 11:59PM. If you will need more time than this to complete your revisions, please reply to this message or contact the journal office at plosone@plos.org . Please include the following items when submitting your revised manuscript:

We look forward to receiving your revised manuscript.

Kind regards,

Tung Truong-Thanh

Academic Editor

PLOS ONE

Journal Requirements:

Additional Editor Comments:

Thank you for submitting the revised version of your manuscript. We appreciate the effort you have made in addressing the reviewers' comments and revising the manuscript. However, we have noted a few minor points that require additional clarification or refinement before final acceptance. Please find the additional reviewer comments below.

Reviewers' comments:

Reviewer's Responses to Questions

**Comments to the Author**

1. If the authors have adequately addressed your comments raised in a previous round of review and you feel that this manuscript is now acceptable for publication, you may indicate that here to bypass the “Comments to the Author” section, enter your conflict of interest statement in the “Confidential to Editor” section, and submit your "Accept" recommendation.

Reviewer #2: All comments have been addressed

Reviewer #3: (No Response)

2. Is the manuscript technically sound, and do the data support the conclusions?

Reviewer #2: Yes

Reviewer #3: (No Response)

3. Has the statistical analysis been performed appropriately and rigorously? 

Reviewer #2: N/A

Reviewer #3: Yes

4. Have the authors made all data underlying the findings in their manuscript fully available?

Reviewer #2: Yes

Reviewer #3: Yes

5. Is the manuscript presented in an intelligible fashion and written in standard English?

Reviewer #2: Yes

Reviewer #3: Yes

6. Review Comments to the Author

Reviewer #2: (No Response)

Reviewer #3: The manuscript focused on using molecular simulations to identify inhibitors of the LasR quorum sensing system in Pseudomonas aeruginosa. Below is a review of the key aspects of the manuscript:

Strengths:

- The study effectively combines molecular docking, ADMET (absorption, distribution, metabolism, excretion, and toxicity) studies, and molecular dynamics simulations, providing a robust framework for identifying potential quorum sensing inhibitors.

- The inclusion of docking poses and protein-ligand interaction diagrams enhances the clarity of the results.

- The manuscript shows evidence of addressing prior reviewer comments, such as improving the introduction, rationale for compound selection, and methodology clarification.

Areas for Improvement:

Introduction and Background:

- While the introduction is well-rounded, it could further emphasize the novelty of focusing exclusively on thiazole derivatives over other compound classes previously reported for quorum sensing inhibition.

- Expand on how this work compares or improves upon prior studies in the field.

Molecular Dynamics Details:

- The manuscript discusses 50 ns molecular dynamics simulations but does not clearly justify why this timespan was chosen or if longer simulations were considered.

- Further discussion on the stability and interactions observed during simulations could be beneficial.

ADMET Analysis:

- While pharmacokinetic and toxicity parameters were considered, it is unclear if experimental validation (e.g., stability in biological fluids) will be pursued in future work.

Conclusions:

- The conclusions could benefit from a stronger emphasis on the implications of this work and a discussion of the next steps, such as experimental validation or broader applications.

Consequently, the study is primarily computational. The study should acknowledge the inherent limitations of in silico approaches, including the reliance on computational predictions that may not fully replicate biological complexities, and emphasizes the need for experimental validation to confirm the efficacy and safety of the identified compounds in real-world biological systems. Authors should expand on molecular dynamics findings and include additional validation metrics if available (e.g., simulation run length should be 100 ns or Molecular Dynamics (MD) Simulations should be performed on 5 compounds). A revised manuscript should diligently comprehensively address the aforementioned concerns.

7. PLOS authors have the option to publish the peer review history of their article (what does this mean? ). If published, this will include your full peer review and any attached files.

**Do you want your identity to be public for this peer review?** For information about this choice, including consent withdrawal, please see our Privacy Policy .

Reviewer #2: No

Reviewer #3: No

---

## [Author Response · Author response to Decision Letter 1]

3 Feb 2025

Comments to Reviewer

A. Introduction and Background:

1. While the introduction is well-rounded, it could further emphasize the novelty of focusing exclusively on thiazole derivatives over other compound classes previously reported for quorum sensing inhibition.

Ans. The detailed explanation has been added on page number 7 (Red coloured text)

2. Expand on how this work compares or improves upon prior studies in the field.

Ans. The detailed explanation has been added on page number 8 (Red coloured text)

B. Molecular Dynamics Details:

- The manuscript discusses 50 ns molecular dynamics simulations but does not clearly justify why this timespan was chosen or if longer simulations were considered.

- Further discussion on the stability and interactions observed during simulations could be beneficial.

Authors are thankful to the reviewer for valuable comment. The 50 ns MD simulation was selected for its ability to capture the conformational stability, dynamic behaviour, and major interactions between the protein and ligand accurately. Simulation lengths of this duration have already been shown in similar studies [1,2] to have been sufficient for the analysis of RMSD, RMSF, hydrogen bonding, and other interaction metrics. Furthermore, 10 ns intervals were selected for the temperature-control simulations to guarantee a good balance between speed and accuracy. The results showed stable trajectories with minor fluctuations in the RMSDs, indicating that the system had reached thermal equilibrium at 300 K. The analyses of hydrogen bonds and protein-ligand contacts showed consistent stability and binding interactions even within this timeframe. Although further simulated trajectories would provide sufficient insights into more infrequent conformational changes or dynamic events, the present findings based on the 50 ns simulation approached the objectives of this study. Future work may involve extended simulations to promote the observation of relatively complex or long-term molecular events.

Furthermore, the discussions regarding the stability and interaction of the system (RMSD, RMSF, hydrogen bonds, and protein-ligand contacts) are already included in detail and supporting scientific explanations at Section 3.5 of Results and Discussion.

References

1.Wang X, Chen Y, Zhang S, Deng JN (2022) Molecular dynamics simulations reveal the selectivity mechanism of structurally similar agonists to TLR7 and TLR8. PLoS ONE 17(4): e0260565. https://doi.org/10.1371/journal.pone.0260565

2.Ghahremanian S, Rashidi MM, Raeisi K, Toghraie D. Molecular dynamics simulation approach for discovering potential inhibitors against SARS-CoV-2: A structural review. J Mol Liq. 2022 May 15;354:118901. doi: 10.1016/j.molliq.2022.118901. Epub 2022 Mar 9. PMID: 35309259; PMCID: PMC8916543.

C. ADMET Analysis:

- While pharmacokinetic and toxicity parameters were considered, it is unclear if experimental validation (e.g., stability in biological fluids) will be pursued in future work.

Authors are thankful for insightful comments. Although the study primarily relies on the in-silico based pharmacokinetic model building and the toxicity evaluation part, experimental validation will serve a crucial role to gain a translational impact on the study. Our future research experimentation will be focussed on analysing screened compounds stability in biologically relevant fluids (e.g., plasma, gastric fluid) using recommended experimental approaches to infer insights of crucial importance on the relevance and robustness of the findings, which would further boost the translational capacity of the same.

Accordingly, we have made justified additions in the manuscript (Conclusion) highlighting this valuable suggestion.

D. Conclusions:

The conclusions could benefit from a stronger emphasis on the implications of this work and a discussion of the next steps, such as experimental validation or broader applications.

Consequently, the study is primarily computational. The study should acknowledge the inherent limitations of in silico approaches, including the reliance on computational predictions that may not fully replicate biological complexities, and emphasizes the need for experimental validation to confirm the efficacy and safety of the identified compounds in real-world biological systems. Authors should expand on molecular dynamics findings and include additional validation metrics if available (e.g., simulation run length should be 100 ns or Molecular Dynamics (MD) Simulations should be performed on 5 compounds). A revised manuscript should diligently comprehensively address the aforementioned concerns.

Authors strongly agree with reviewer’s comment. Accordingly, justified additions are made in the conclusion section of the manuscript (highlighted in red).

---

## [Decision Letter · Decision Letter 2]

26 Feb 2025

Molecular simulation-based investigation of thiazole derivatives as potential LasR inhibitors of Pseudomonas aeruginosa

PONE-D-24-31938R2

Dear Dr. Kausar,

We’re pleased to inform you that your manuscript has been judged scientifically suitable for publication and will be formally accepted for publication once it meets all outstanding technical requirements.

Kind regards,

Tung Truong-Thanh

Academic Editor

PLOS ONE

Additional Editor Comments (optional):

Reviewers' comments:

Reviewer's Responses to Questions

**Comments to the Author**

1. If the authors have adequately addressed your comments raised in a previous round of review and you feel that this manuscript is now acceptable for publication, you may indicate that here to bypass the “Comments to the Author” section, enter your conflict of interest statement in the “Confidential to Editor” section, and submit your "Accept" recommendation.

Reviewer #3: All comments have been addressed

2. Is the manuscript technically sound, and do the data support the conclusions?

Reviewer #3: Yes

3. Has the statistical analysis been performed appropriately and rigorously? 

Reviewer #3: Yes

4. Have the authors made all data underlying the findings in their manuscript fully available?

Reviewer #3: Yes

5. Is the manuscript presented in an intelligible fashion and written in standard English?

Reviewer #3: Yes

6. Review Comments to the Author

Reviewer #3: The authors have revised or provided comprehensive feedback on this manuscript. It is recommended that the author carefully review the text for spelling errors, particularly the substance name in Figure 1.

7. PLOS authors have the option to publish the peer review history of their article (what does this mean? ). If published, this will include your full peer review and any attached files.

**Do you want your identity to be public for this peer review?** For information about this choice, including consent withdrawal, please see our Privacy Policy .

Reviewer #3: No

---

## [Editor Report · Acceptance letter]

PONE-D-24-31938R2

PLOS ONE

Dear Dr. Kausar,

I'm pleased to inform you that your manuscript has been deemed suitable for publication in PLOS ONE. Congratulations! Your manuscript is now being handed over to our production team.

Kind regards,

on behalf of

Dr. Tung Truong-Thanh

Academic Editor

PLOS ONE